# Techno-Concepts for the Cultural Field: n-Dimensional Space and Its Conceptual Constellation

**Nuria Rodríguez-Ortega** 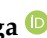

iArtHis_Lab Research Group, Art History Department, Universidad de Málaga, 2971 Málaga, Spain; nro@uma.es

**Abstract:** This paper advocates an epistemological turn in the field of digital art history and cultural heritage studies. This epistemological turn is understood as the elaboration of a new (or renewed) epistemic apparatus that allows us to understand and interpret cultural phenomena from the perspective of a different order of thought. This epistemological labor is conceived of as an «epistemological technical practice», which means integrating epistemological reflection and production into technical making and design. Within this framework of discussion, this paper introduces the idea of a techno-concept, which is defined as a co-production between the machine rationale and the human thought/imagination. As particular cases, this paper argues that the mathematical concepts of n-dimensional, vector and latent spaces constitute examples of techno-concepts that can be reappropriated and reworked for cultural analysis and interpretation. This paper offers a preliminary inquiry, in which certain epistemological propositions are exposed as open gates for further discussions.

**Keywords:** techno-concept; n-dimensional space; latent space; vector space; epistemological turn; digital art history

## 1. Epistemological Turn and Techno-Concepts for the Cultural Field

### 1.1. Advocating an Epistemological Turn in the Field of Digital Art History

The progressive incorporation of digital technologies and computational methods in art history and culture research that has been taking place for some decades has crystallized in a large number of projects that, in different ways, have produced computational and digital devices, artifacts, objects, resources and infrastructures, as well as having amplified the research perspectives on certain themes inscribed in the intellectual concerns that have traditionally occupied art history and cultural heritage studies [1–5] However, epistemological production itself—understood here as the explicit elaboration (or re-elaboration) of concepts, categories of analysis and theoretical models that allow us to understand and interpret cultural phenomena from a different order of thought—has not yet occurred regularly and systematically. According to my view, one of the research priorities we should address in the coming years is precisely the configuration of a new (or renewed) epistemic apparatus so that, following an iterative process, the production of digital «artifacts» is accompanied by a conceptual and theoretical production

This paper is concerned with this purpose. Its objective is, therefore, to contribute to the reflection of the concepts on which attention should be focused and how they could be used as categories of cultural analysis and interpretation. It is important to say that I understand this epistemological labor not only as a merely intellectual production, but as an epistemic making inseparable from technical practices so that epistemological production becomes an «epistemological technical practice» [6]. Epistemological technical practice means integrating epistemological reflection and production into the technical making and design and using technical making and design to elaborate a renewed theoretical body of concepts.

### 1.2. Techno-Concepts for the Cultural Field

Given this framework of reflection, in this paper, I wish to introduce the idea of a techno-concept, which constitutes the essential pillars, the foundational conceptual layer, of a broader research project that will be developed in the coming years. The idea of a techno-concept suggests, in the first place, that the configuration of this new epistemology that we need does not merely rest on an exercise of ex novo invention (perhaps this applies to any epistemological formulation); rather, it is about investigating how concepts belonging to the tradition of philosophical thought, cultural theory and art history can be reformulated (or reinvented) in the context of contemporary techno-mediation processes. That is why, following the suggestion of Javier Echevarría and Lola Almendros [7], I propose to name these concepts with the term 'techno-concepts'. The prefix 'techno-' has a double meaning: first, techno-concepts represent techno-scientific «entities» in the sense that only in or through techno-scientific procedure do they have intellectual existence; and second, techno-concepts, in their notional formulation, constitute a co-production between the techno-scientific rationale (i.e., algorithmic or computational) and human thought/imagination. Thus, as I try to argue in this paper, the identity and characteristics of a particular techno-concept are presented to us as a co-elaboration between the operational logic of the technology that produces it and the human cognition that appropriates it in the process of understanding and interpreting it. However, since the idea of a techno-concept cannot be separated from the technology that produces it, the epistemological assumptions embedded in the process of technological production almost irremediably shape the interpretation and conceptual elaboration carried out by the human subject. Consequently, a techno-concept forces us to be in a state of continuous tension; it forces us to swing (as we will see throughout this text) between the possibilities of narrative and epistemological broadening that it promises and, at the same time, the conceptual determinations, reductionisms and limits that might imply technological conditioning. In these ways a techno-concept generates its own problematization field or sets of problems. Offert and Bell [8] have argued the existence of an epistemological split between computer scientists and engineers and digital humanities when approaching critical problems in the field of machine learning. I consider that the notion of a techno-concept could be an intellectual instrument to overcome this split, although this thought would need further examination.

In its second meaning, a techno-concept involves a post-human perspective of analysis in the sense that it entails the awareness that human beings cease to be the sole origin of the production of knowledge, and therefore, postulates the need to inquire into the emerging forms of co-production as a way to overcome the epistemologically reductive character of anthropocentrism. Thus, the idea of a techno-concept also constitutes a way of exploring human decentering from the interpretative and explanatory processes of the phenomena that surround us, including the subject itself, understanding «decentering» here as the recognition that there are behavioral logics that escape human intelligibility (that is, that we are not capable of understanding with our current mental and cognitive equipment) while having a rationale of their own.

### 1.3. n-Dimensional Space and Its Associated Concepts as Study Case

In this paper, I focus on the concept of n-dimensional space, and, more specifically, on what is called high-dimensional vector space and on some of its associated concepts, such as a vector or latent space. In his famous 1984 novel *Neuromancer*, William Gibson, coiner of the term 'cyberspace', already suggested that this new space (or techno-space) would not be so much a sort of virtual reality but a vector space, as Pasquinelli and Joler [9] noticed. I humbly believe that Gibson could not be more right.

The current relevance of the concept of n-dimensional space for cultural analysis and interpretation must be related, in the first place, to the «ontological» transformation that cultural objects have been undergoing for decades as a result of their digitization or direct digital production: cultural objects, regardless of their nature (images, words, sounds, etc.), in their digital mode of existence are essentially matrices or sets of numerical

data. It is precisely this transformation that makes their computation possible. From a computational point of view, images or texts are nothing more than a spatial surface of numerical information from which it is possible to extract (also) numerical characteristics using computational systems. Therefore, this ontological transformation also implies an epistemological transformation insofar as cultural objects that are transformed into digital forms become a problem of a computational and mathematical order.

The numerical condition that now defines the existence of digital cultural objects complementarily entails a transformation of the concept of space, understood here as the place in which cultural contents live, manifest or unfold. It is in relation to this fact where the concept of a high-dimensional space finds its relevance since, strictly speaking, it can be said that n-dimensional space is where cultural objects live and exist today; that is, as I will try to argue in the following section, cultural objects today «are» n-dimensional spaces that are produced and can be explored using computational methodologies.

This exploration has been carried out for a long time in the field of computational linguistics, NLP, computer vision and, in general, in the broad field of what we now call the computational humanities. Over the last few years, these computational methodologies have been shown to be enormously fruitful in relation to different aspects of humanistic and cultural research, which have been expanded and enriched. Today, the literature on the subject is immense, which is a clear symptom of the explosion experienced in this research field over the last decade [10–14]. However, to date little attention has been paid to these concepts as the new cultural objects that they are. Generally, these concepts are taken for granted, as the focus of interest is on the research results and the analytical possibilities that these technologies open up, or on the discussions of the critical issues that AI practices and their outputs involve. Notwithstanding the foregoing, the contributions by Pasquinelli and Joler [9], Offert [15,16] and Offert and Bell [17] deserve to be highlighted. Therefore, rather than examining their analytical possibilities, I will instead focus on some of the conceptual and epistemological implications that are embedded in the concepts of n-dimensional, vector and latent spaces. It should be kept in mind that the concepts under examination are at the same time techno-concepts, understood as intellectual entities; and techno-objects, understood as concrete and particular materialities and devices with which we interact. These techno-objects play, then, an essential role in our ways of understanding and thinking, and, consequently, in the reshaping of the associated techno-concepts through an interesting (and perverse) feedback loop.

It must be stated that what follows is a preliminary approach that will require further and systematic examination. The concepts (or techno-concepts) are presented here briefly, as opening gateways for deeper investigations rather than conclusive ideas.

## 2. n-Dimensional, Vector and Latent Spaces

n-Dimensional space is an abstract mathematical concept that, as its name indicates, conceptualizes a space that is configured by a multiple number of dimensions. The idea of n-dimensional space has no formal or empirical correlate; it is rather an intellectual generalization that is based on the logical possibility of multidimensional objects and serves the purpose of formalizing, on a common foundation, different ideas of space, geometric space and, therefore, different types of geometries. For example, regarding the influence of the concept of n-dimensional space on Ernst Cassirer's and Aby Warburg's concept of *Denkraum*, see [18].

This general (and qualitative) idea of n-dimensional space can be materialized in certain mathematical models that, in turn, can be operationalized and computed. This is the case, for example, of n-dimensional vector space, which can be informally defined as a space of n coordinates where numerical information is distributed, with each coordinate representing a dimension or characteristic of the domain being analyzed. Consequently, a n-dimensional vector space allows us to represent multidimensional objects as commensurable entities [15] and to analyze them in their multiple dimensions. This type of mathematical space, although certainly counterintuitive for human cognition, entails

undoubted epistemological and research potentialities by increasing the possibilities of analyzing complex objects and phenomena. However, it also entails critical considerations when the domains of analysis and representation are cultural and social ones.

### 2.1. How do Cultural Objects Become High-Dimensional Vector Spaces?

Before continuing, a question arises: How do cultural objects become n-dimensional spaces or (also called) high-dimensional spaces? Let us think of a digital image (Figure 1). A digital image is simply a big collection of pixels, with each pixel representing a value (color, intensity, etc.) within some range. Each pixel is, thus, a dimension that characterizes, in quantitative terms, the image. A low-resolution image might be 32-by-32 pixels, which results in a 1024-pixel image, or an image made up of 1024 dimensions. In other words, the digitized image of our example constitutes a 1024-dimensional space. In the case of color images, this size would have to be multiplied by the three color channels (RGB), which would result in a 3072-dimensional space.

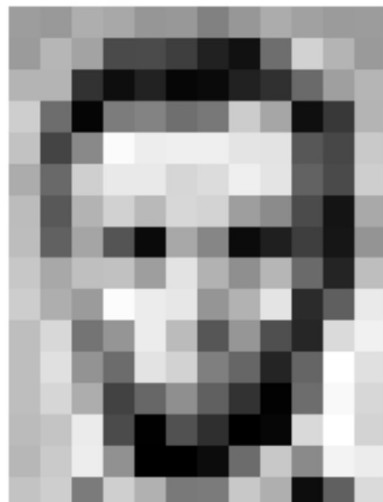
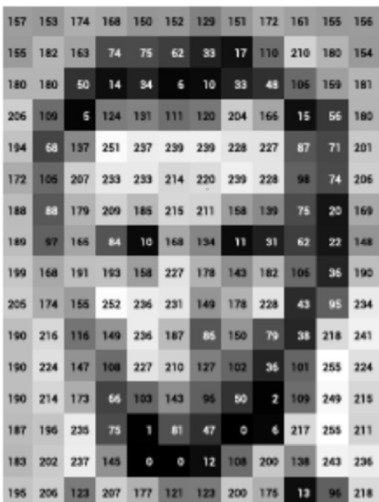
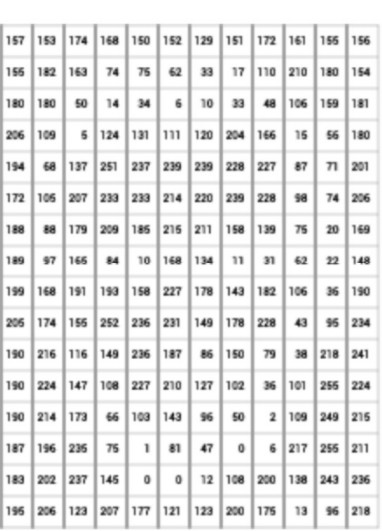

**Figure 1.** Digital image as matrix of numerical values. Numerical values represent pixel's brightness in this example. Golan Levin. «Image Processing and Computer Vision», ofBook (work in progress). Available on: https://openframeworks.cc/ofBook/chapters/image_processing_computer_vision.html (accessed on 28 July 2022).

The same happens when considering textual or linguistic corpora. Texts can be thought as unstructured datasets of high dimensionality. When we work with texts or documents, we are working with sets of words or terms. Each word in a document is a dimension that characterizes that document in some way. Moreover, each word-dimension can be translated into a number according to different measures and values. In the example shown in Table 1, a matrix is depicted in which the terms are quantified according to their TF-IDF values. TF-IDF is a measure that quantifies the frequency of occurrence of a word (or token) in inverse proportion to the number of documents in which it appears in a corpus. The result is a 10-dimensional space.

**Table 1.** Simplified representation of textual corpus dimensionality.

|       | Beauty | Genius | Form   | Line   | Drawing | Painting | Space  | Pattern | Surface | Plane  |
|-------|--------|--------|--------|--------|---------|----------|--------|---------|---------|--------|
| Doc 1 |        | 0.0811 | 0.1899 | 0.1722 |         |          | 0.0893 |         |         | 0.9922 |
| Doc 2 |        | 0.0681 | 0.1787 |        |         | 0.2942   |        | 0.2942  | 0.2956  |        |
| Doc 3 | 0.3197 |        |        | 0.1911 | 0.3298  |          |        |         |         | 0.1922 |

In the example shown in Figure 2, terms are quantified according to their degree of proximity to each of the four dimensions under analysis. Thus, this microcorpus of nine words generates a four-dimensional space. Naturally, when considering real projects, the number of dimensions increases dramatically to hundreds of thousands or millions.

**Dimensions**

| Word vectors | | | | | |
|---|---|---|---|---|---|
| dog | -0.4 | 0.37 | 0.02 | -0.34 | ■ animal |
| cat | -0.15 | -0.02 | -0.23 | -0.23 | ■ domesticated |
| lion | 0.19 | -0.4 | 0.35 | -0.48 | ■ pet |
| tiger | -0.08 | 0.31 | 0.56 | 0.07 | ■ fluffy |
| elephant | -0.04 | -0.09 | 0.11 | -0.06 | |
| cheetah | 0.27 | -0.28 | -0.2 | -0.43 | |
| monkey | -0.02 | -0.67 | -0.21 | -0.48 | |
| rabbit | -0.04 | -0.3 | -0.18 | -0.47 | |
| mouse | 0.09 | -0.46 | -0.35 | -0.24 | |
| rat | 0.21 | -0.48 | -0.56 | -0.37 | |

**Figure 2.** Simplified representation of word vectors. Jayesh Bapu Ahire. Introduction to Word Vectors. Available on: https://dzone.com/articles/introduction-to-word-vectors (accessed on 28 July 2022).

In a high-dimensional space, each of the cultural elements (whether they are images, documents, words, sounds, etc.) is encoded in a vector of characteristics or (numerical) dimensions. Thus, the resulting vector, as an ordered tuple of values, stands as a (numerical) representation of the cultural object. Coming back to our toy examples, observe that document 3 in Table 1 is represented by the vector [0.3197, 0.1911, 0.3298, 0.1922], the word 'dog' in Figure 2 by the vector [−0.4, 0.37, 0.02, −0.34], and the image would also be represented by a unique vector of 1024 dimensions. Therefore, in a high-dimensional space, cultural objects exist as vectors (or vectorial representations) that are projected in different directions of the high-dimensional space and are distributed in certain positions according to the encoded characteristics (Figure 3).

### 2.2. ML, Deep Learning and the Emergence of a Vectorized Culture

It must be said, however, that the increasing relevance of high-dimensional vector spaces in the field of cultural research is also directly associated with the expansion of deep learning technologies for linguistic and visual analysis since these neural networks necessarily require that the information to be computationally analyzed is transformed into vectors. Thus, in the field of deep learning, vectorization technologies have been developed to produce vector space models. Consequently, these high-dimensional vector space models are built according to the algorithmic logics that define the functioning of AI and then incorporate the whole set of epistemological assumptions that are embedded in such logics. Due to their current relevance for cultural analysis, it is crucial to understand, albeit summarily, the logics of these high-dimension vector space models.

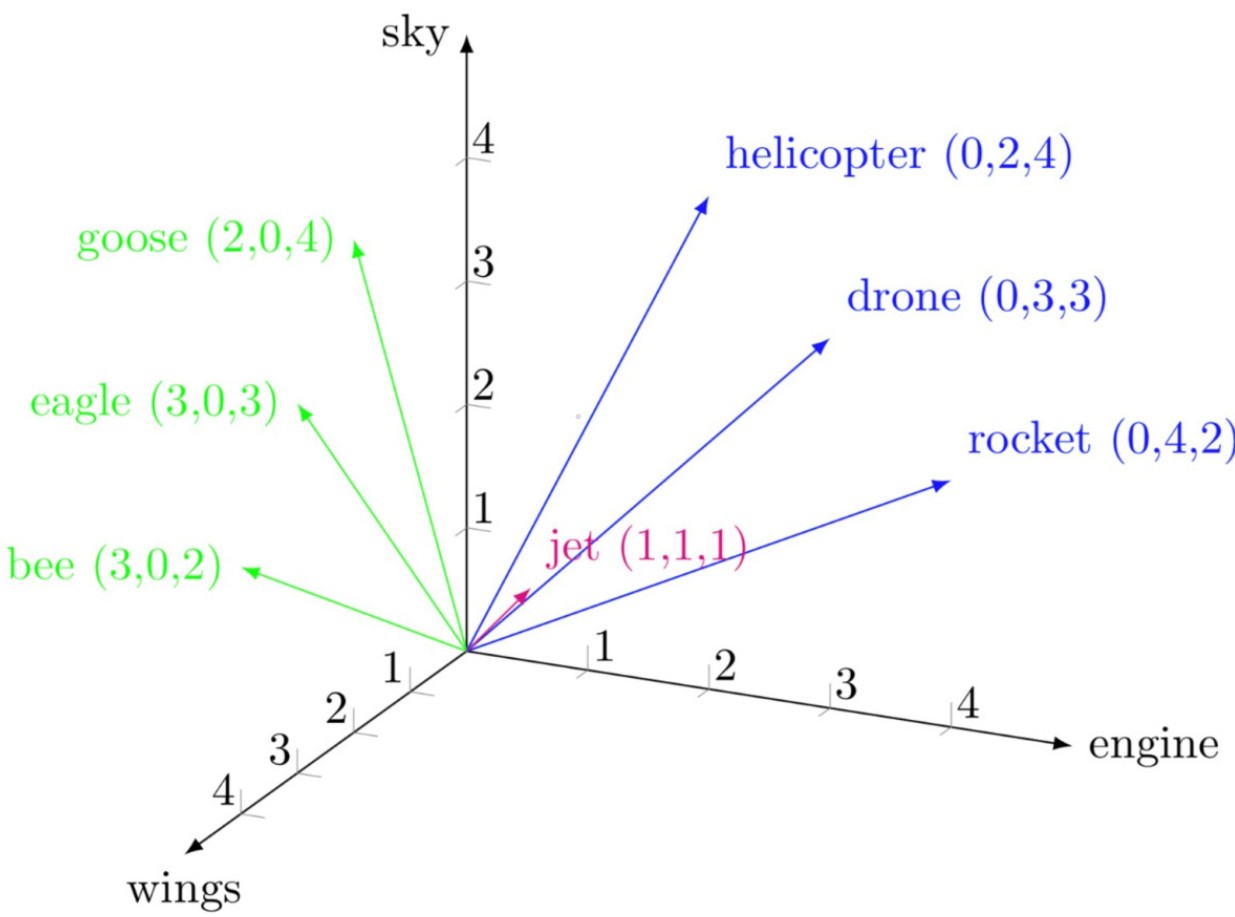

**Figure 3.** Vector space of seven words in three contexts. Guillaume Desagulier. Word embeddings: the (very) basics. Available on: https://corpling.hypotheses.org/495 (accessed on 28 July 2022).

Despite the variety of machine learning applications, all machine learning logic is based on the same set of operations: taking the set of instances or data of the training dataset as a base, machine learning technologies recognize and extract the statistical distribution of patterns that define the (non-directly observed) dataset structure. This statistical distribution of patterns constitutes a statistical model that stands as a representation of the corpus (or domain of reality) under analysis. In order to optimize their performing, in their hidden layers deep neural networks encode high-dimensional spaces into what are known low-dimensional latent spaces, that can be informally defined as an internal and compressed representation of all relevant features of the dataset. In latent spaces, similar data are closer together, therefore, as we see later, in latent spaces similarity is understood as the spatially measured relationship between the feature vectors encoded by the network. However, what I want to highlight for now is the condition of latent spaces as models that encode a representation of the cultural domain under analysis as a whole, not only of an individual element. It can be said that this space, which is based on a quantitative spatial representation, is the way a computational system understands the entire domain. Certainly, the encoded representation of real-world depends on the statistical approach (discriminative or generative) employed for modeling, thus, meanwhile, discriminative approaches only use the most relevant features to perform optimal classifications, generative approaches learn the complete distribution of features attempting to obtain a model of the world as it is [8], but in all cases we handle with «a» representation of the entire cultural domain.

High-dimensional vector spaces and, particularly, latent spaces have been shown to be very useful in understanding the non-directly observable characteristics and structures of certain domains. Likewise, vector space models generated by AI are used to perform a

diverse range of tasks (data classification, image/text generation and reconstruction, etc.). That is why vector space models are, at the same time, computational models (representations) that support algorithmic-based methods for the inductive discovery of meaningful structures or the generation of «new» entities or signs. Both facets (computational model and exploration-generation tool) must be taken into account when addressing their epistemological and critical implications.

As I pointed out earlier, making a mental image of a high-dimensional space is very complicated since it is practically impossible to process using our human brain, which is used to thinking—at least until recently—in four-dimensional parameters. The counterintuitive character of these vector spaces lies not only in the number of dimensions but also in some of their properties and in the algorithmic rationale that operates on and produces them, which are sometimes unintelligible to our forms of understanding. One of the ways the structure of the vector space can be made intelligible to us is numerically, for example, as a ranking of the nearest data points in the vector space (Table 2 [19]).

**Table 2.** Topics in the annotations of Magritte's artworks and the top 10 nearest descriptors. J.F. Chartier et al. A Data-Driven Computational Semiotics: The Semantic Vector Space of Magritte's Artworks. *Semiotica* **2018**, *2019*(230), 19–69 (31, fragment).

| Topic #4 EYE | cos | Topic #5 RIDDING HORSE | cos | Topic #6 SEA | cos | Topic #7 PIPE | cos |
|---|---|---|---|---|---|---|---|
| <iris_(eye)> | 0.61 | <horse> | 0.59 | <sea> | 0.62 | <steam_(pipe)> | 0.81 |
| <pupil> | 0.60 | <horse_riding> | 0.56 | <wave> | 0.60 | <shank_(pipe)> | 0.81 |
| <white_of_the_eye> | 0.60 | <rein> | 0.54 | <water> | 0.59 | <head_(pipe)> | 0.81 |
| <eyelash> | 0.53 | <flange> | 0.54 | <scum> | 0.55 | <heel_(pipe)> | 0.81 |
| <perl> | 0.51 | <bit> | 0.54 | <skyline> | 0.47 | <bowl_(pipe)> | 0.81 |
| <lacrymal_caruncle> | 0.51 | <tail_(horse)> | 0.51 | <sky> | 0.47 | <lop_(pipe)> | 0.80 |
| <eye_ball> | 0.49 | <crop> | 0.51 | <beach> | 0.44 | <pipe> | 0.77 |
| <necklace> | 0.47 | <hoof> | 0.49 | <swell> | 0.40 | <plate> | 0.67 |
| <support> | 0.47 | < saddle_(horse)> | 0.49 | <surge> | 0.37 | <inscription> | 0.24 |
| <lips> | 0.47 | <mane_(horse)> | 0.49 | <cloud> | 0.37 | <tobacco> | 0.22 |

However, vector spaces undoubtedly become intelligible in a more intuitive way when the spatial metaphor is kept, that is, when vector spaces are visually displayed as a spatial form or structure. For this purpose, specific dimensionality reduction algorithms are applied. These algorithms try to maintain the structure of the data points in the high-dimensional space while projecting it in two and three dimensions so that we can actually visualize it. In this way, the visualizations generated (see, for examples, Figures 4 and 5) provide us with a more or less approximate image of how data points are distributed in the high-dimensional space, which we cannot humanly see and, at best, find it very difficult to think about. We will return to these visualizations in the next section.

### 2.3. Distorted or Augmented Gaze: Disparate Approaches to Vectorization Processes

The way of approaching and interpreting the logics behind the generation of vector space models is not homogeneous. For example, Pasquinelli and Joler [9] (among others) describe them as a large lens that diffracts reality: reality (social, cultural, physical, etc.) that is transformed into data (which in itself implies a certain reconfiguration of the reality to which it refers) is compressed by the algorithms and diffracted into the world through the lens which is the statistical model (or internal representation). What machine learning technologies produce, then, is compression and distortion of the flow of information whose central component is just the statistical model that diffracts perception. Therefore, for Pasquinelli and Joler, what machine learning produces is, in reality, a diffracted rationality and a «statistical hallucination». Furthermore, it must be considered that ML can only operate within the framework constituted by the training dataset. It cannot recognize anything that has not been included in the dataset, nor can it generate anything new beyond the dataset and/or information whit which it has been trained t. That informational framework constitutes the cognitive edge and boundary of ML. This circumstance, together with the fact that computational methods introduce their own epistemic logics in the process of rebuilding the real-world in «a» certain model instantiated in the latent vector

space, leads us to say that what is produced is a virtual reality only existing within the framework of the specific computational method used (see Offert [16] for the particular case of GAN).

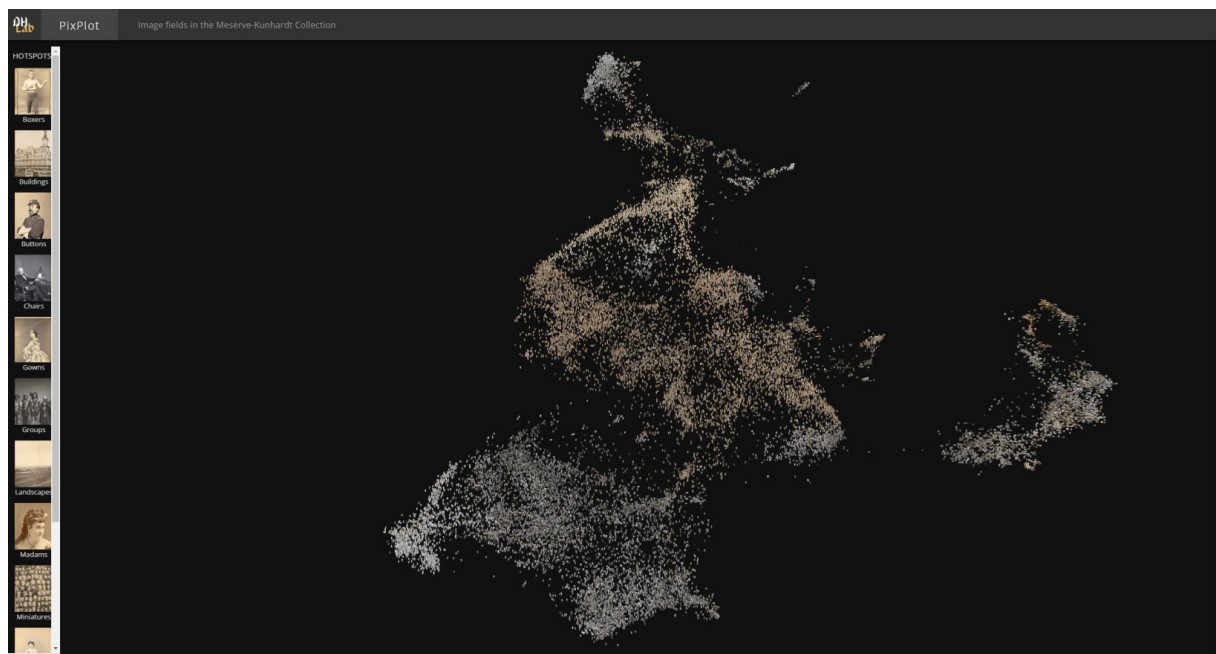

**Figure 4.** Two-dimensional manifold of image vector space with UMAP (Meserve Kunthardt Collection). Available on: https://s3-us-west-2.amazonaws.com/lab-apps/pix-plot/index.html (accessed: 28 July 2022) (screenshot).

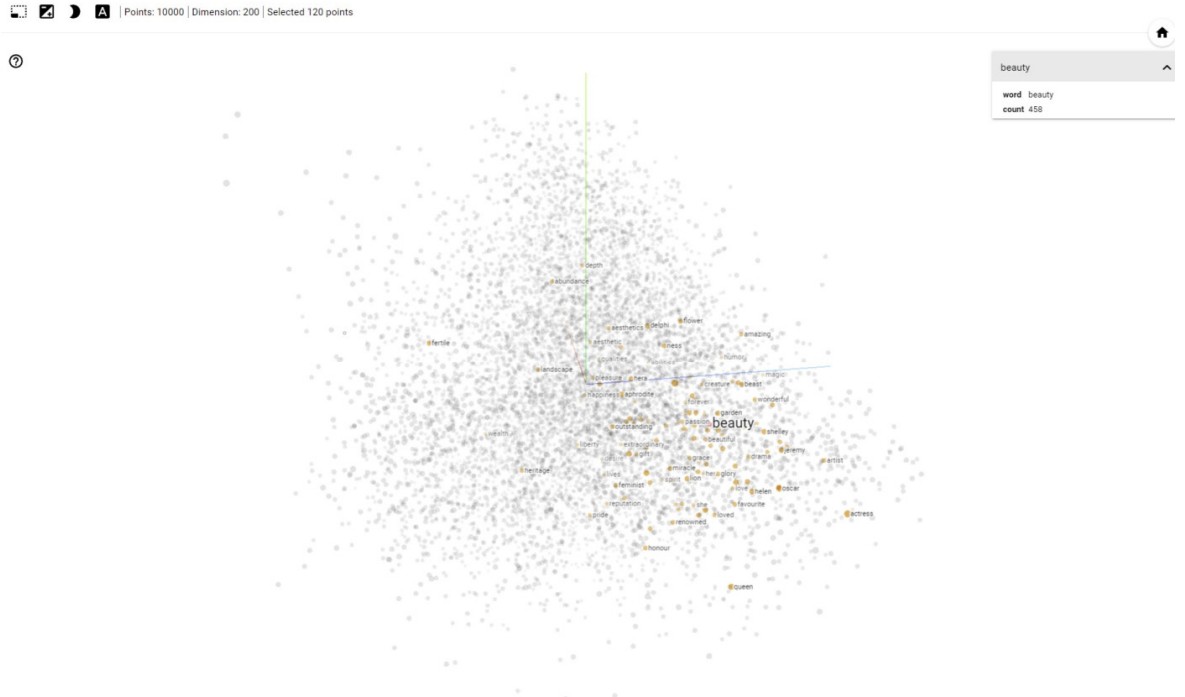

**Figure 5.** Three-dimensional embedding of word vectors space with PCA (cosine similarity distance). Available on: https://projector.tensorflow.org/ (accessed: 28 July 2022) (screenshot).

The opposite approach is represented by those positions that understand machine learning as a method that provide us with augmented and magnified knowledge of the reality we observe due to their capacity to make intelligible implicit characteristics, patterns and correlations across a huge space of data. This is, for example, the idea behind the concept of cultural analytics that has been elaborated, theorized and put into practice by Lev Manovich [20]. Although Manovich's methods are primarily focused on image processing by means of techniques based on the extraction of their low-level features, the arguments in favor of their employment for cultural analysis and interpretations are the same. Manovich himself defined data as a medium that translates reality through a process in which only certain characteristics are captured and encoded as data. As such a medium, data therefore have capabilities but also limitations. However, based on an antithetical approach between the discretizing–bounding function of the logical–linguistic categories and the continuous–gradational nature of numerical ranges, Manovich concludes that numerical data constitute a better representation of cultural reality that is essentially analogical. Consequently, numerical data analytics becomes a better instrument for analyzing its dimensions. I believe that, here, Manovich falls into the epistemological fallacy of considering that there are, in themselves, ways of analyzing and representing reality that are more correct than others. The continuous–gradational nature of numerical ranges brings undoubted analytical and interpretative benefits, as long as it is considered one way (among others) of interrogating cultural reality with its own limitations. When Manovich says that lexicalizations used in the classification of images do not reflect the characteristics of visual language maybe he does not take into account that they do reflect other important cultural issues that could even be more relevant for certain research questions.

In this regard, my position is based on a strategic and pragmatic stance. I consider that a scenario that is defined by the ubiquity and prevalence of AI compels us to seek productive frameworks to expand our comprehension of cultural phenomena and processes. I consider it essential to explore the possibilities of «seeing» that AI offers while bearing in mind that it is a diffracted vision. In short, I think it is more interesting to think that AI helps us to see differently than to think that it helps us to see more and better. It is in this different seeing where, in my view, lies the heuristic and epistemological potential of these technologies.

## 3. Problematization Fields and Epistemological Implications

The problematization fields to which I refer below are not exhaustive; they have been selected based on those that, for the moment, I consider to require more attention, although I am aware that it may be a very subjective decision.

### 3.1. Vectorized Culture and Cultural Vectorization

The phenomenon of «cultural vectorization» (and its associated concepts and practices) has interesting epistemological, critical and narrative implications, as will be argued later. However, it is the transformation of the representational order of cultural heritage that it is producing what must be addressed in the very first place. As aforementioned, cultural objects (whether words, images, texts, sounds, etc.) are no longer only rematerialized in numerical information but also encoded in vectors, which thus become the new representational entities of cultural heritage that, therefore, remains numerically encoded according to certain characteristics or values. Likewise, the high-dimensional vector space models produced using deep learning neural networks constitute a vectorization of the entire cultural domain under analysis. In other words, the totality of the vectors represents the totality of the structure of the dataset such that the vector space stands as «a» model that represents and describes the cultural domain in its entirety.

If we take into account the fact that vectorization processes are highly increasingly common phenomena that result from the growing use of deep learning technologies in linguistic and visual analysis, it can be affirmed that we are witnessing a transformation in how cultural heritage is recorded, preserved and transmitted and, with that, the set

of knowledge, beliefs, values and ideas that cultural objects convey. In a strict sense, it could be said that vectors and high-dimensional vector spaces are being instituted (in our computational society shaped by AI) as the new models for recording and ordering the memory of culture, thus displacing what, until now, were considered the fundamental models: the book, archive and database artifacts.

Simultaneously, a reconfiguration of the concept and meaning of what a cultural object is and of the nature of the conceptual categories used for its description, ordering and interpretation is taking place. If cultural objects are translated into strictly commensurable values, the primary question that arises is about the limits of the commensurability since it is one of the epistemological assumptions embedded in computational approaches. That is, we need to discuss whether it is epistemologically possible that all the dimensions and characteristics of cultural objects can be translated into numbers and codified in vectors, or instead there is some margin for the incommensurable, that is, for what cannot be numerically codified. Even if being epistemologically feasible, this scenario also prompts us to inquire what kind of reductionism would imply the prevalence of a vectorized culture.

Likewise, and given that the logic underneath AI applications can be understood as a process of extraction and production of statistical distribution of patterns, the concept of pattern (or, better said, of techno-pattern, as produced by AI) becomes a central issue that demands an examination of what implies to instantiate cultural objects, processes and phenomena in sets of techno-patterns; what it means to understand cultural domains as sets of techno-patterns correlated in some way; what epistemological assumptions are embedded and how they determine research and interpretation modalities; what convergences and divergences emerge when this idea of a tecno-pattern is put in relation to other concepts of cultural patterns formulated over the past.

A modification in the nature of the analytical and interpretative categories hitherto used is also in fieri. As explained above, a techno-pattern is a distribution of patterns that constitute a model. Consequently, art-historical concepts such as style, genre, period or authorship are no longer intellectual elaborations but statistical models of characteristics in the domain of computational art history. For the same reason, from an algorithmic point of view, classifying or giving order to cultural production is to determine whether a numerical data structure is more or less similar to the pattern encoded in a model. Within this framework, there are no longer cultural objects that «are» baroque (just to put an example), but objects (as a set of characteristics) more or less close to a pattern which has been generated by statistical induction. These classificatory or ordering logics need to be discussed in relation with the empirical-based classificatory methods, also known as inductive classification, phenomenal classification or extensional classification, as well as with the Eleanor Rosch's prototype theory and its further developments. This question, however, falls outside the scope of this paper. So logical categories are replaced by probabilistic ranges. These changes in the nature of objects and conceptual categories entail interesting epistemological derivations, as we will see as follows.

### 3.2. Spatiality: Geometry, Form and Topology

If the concept of an n-dimensional space implies the problematization and reconsideration of some kind of concept, it is precisely the notion of «space» itself. In this sense, I believe that the toy examples of the previous section, although simple, adequately illustrate the fact that these spaces do not act as containers or supports in the traditional sense of the term—that is, spaces where cultural objects, visual contents or writing signs are located, arranged or situated—but are generated by them once they have been transformed into numerical information. Sign and space constitute an indissoluble unity: space is sign and sign is space, and this lack of distinction shifts us from the traditional concept of Euclidean space—according to which space is a pre-existing category, different from what is located in it (it is another dimension, extrinsic to the object)—to the topological concept of space, according to which space is an a posteriori that is generated by and imbricated in the objects themselves.

This shift involves a theoretical perspective that addresses the notion of space as a dimension intrinsically implicated in the definition and production of cultural objects. It also implies bringing back to the center of the debate the notion of spatiality as a category for analysis and interpretation and, more specifically, the «reincorporation» into the cultural and humanistic epistemology of concepts taken from geometry and topology. Neither geometry nor topology constitutes new conceptual frameworks in the field of cultural and humanistic interpretation. In fact, the very conceptualization of the contemporary world as a system of distributed nodes connected by networks, ontologically constituted as a continuity of interweaved and entangled heterogeneous entities, which is moreover in a state of continuous transformation, has favored the new impulse that topology has acquired in recent years as a cultural category [21].

However, in the context of the high-dimensional vector spaces produced and explored by AI technologies, geometry and topology, in addition to providing us with a theoretical and conceptual framework to interpret sociocultural phenomena (which have become topological), also offer us specific mathematical instruments to analyze the geometric and topological structures of data in high-dimensional vector space, whose results, moreover, are visually materialized in certain spatial configurations that facilitate its intelligibility. Before continuing, it is important to notice that, although this paper focuses on high-dimensional vectors spaces as topological and geometrical structures, they can also be algebraic structures.

Tthe spatial distribution of images and works that we observe, for example, in Figures 4 and 5, although reduced to two or three dimensions, responds to the structure that emanates from the computationally established positions and distances of visual and/or linguistic data that are translated into numbers in a vector space. It is just this spatial structure that is meaningful since the implicit knowledge extracted from computation (statistical distribution of patterns) is expressed in the vector space itself (in its configuration and structure). In other words, the sense and meaning of the cultural domain under analysis are codified in terms of distances and spatial structures (or formations). It can be said, then, that a topological–morphological order comes into play since it is precisely the form and spatial structure resulting from data computation that are the fundamental parameters of interpretation.

Thus, geometry and topology, beyond providing us with a theoretical or conceptual framework to think about cultural objects and phenomena, become a device (in Foucault's sense) or artifact, material and concrete, that is generated by the technology itself. It can be said, then, that our spatio-topological imagination is mediated and somehow produced by the analytical methodologies that govern the operating logics of AI, by its epistemological assumptions and, especially, by the indispensable visualizations that we need to make intelligible the structures of high-dimensional vector spaces. This is why I consider we can speak in terms of a techno-spatio-topology and affirm that a spatio-topology, in this framework of computational exploration, becomes a techno-concept and a techno-object. This circumstance entails an approach to geometric and topological concepts from the double perspective of a techno-concept as both a possibility of opening and widening the epistemological and critical horizon for the analysis and interpretation of cultural phenomena and processes, as well as a problem that, as a mode of cultural analysis and interpretation, must be critically discussed to become aware of how these spatio-topological devices determine and model our imagination and, therefore, the narratives that may be derived from them.

Consider, for example, the concept of distance, which is a key concept in cultural interpretation that has been the subject of different conceptualizations and definitions over time. In a vector space, distance becomes a mathematical distance. This issue is well illustrated by the so-called cosine similarity, which is one of the basic operations taken from linear algebra that consists of calculating the similarity between two vectors by taking, as a measure, the cosine of the angle between them (Figure 6).

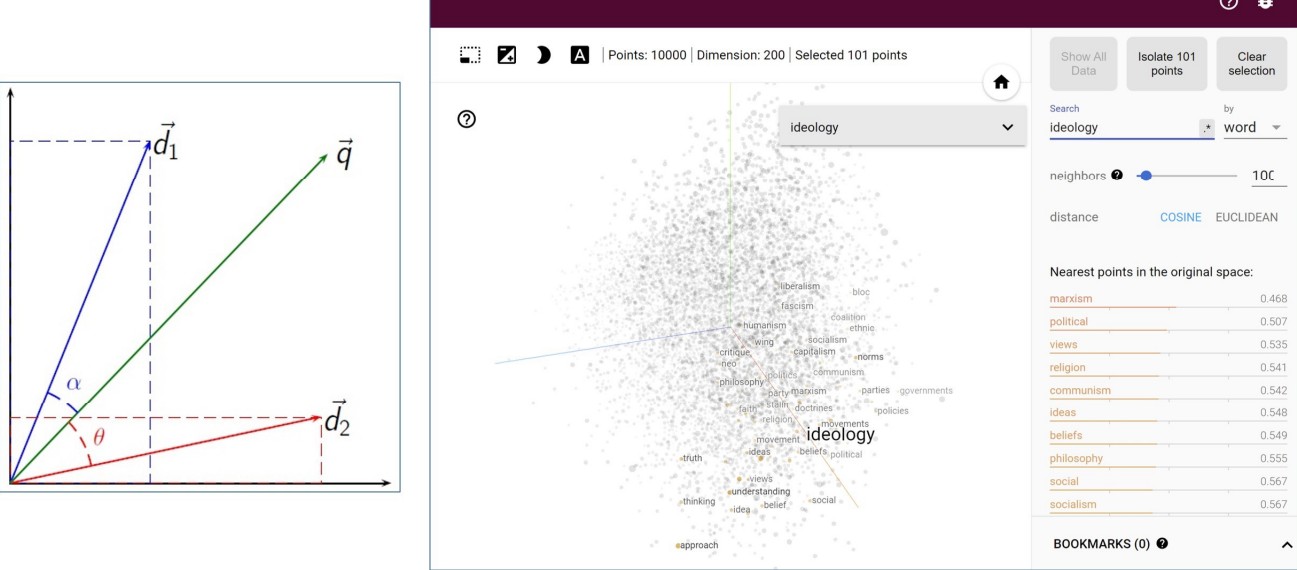

**Figure 6. Left**: cosine similarity function geometrically represented. **Right**: cosine similarity in word vector space showing closest words to "ideology". Available on: https://projector.tensorflow.org/ (accessed on 28 July 2022) (screenshot).

The mathematical interpretation is that the smaller the cosine angle between two vectors, the greater the similarity between these vectors within the vector space. This calculation is used to detect degrees of similarity between cultural elements (transformed into vectors) and to build large-scale cartographies of cultural domains. These large-scale cartographies foster modes of interpretations of cultural domains (visual or linguistic) that take as primary basis their arrangement into groups of similar elements, which in turn underpins modes of narrative based on contiguity and resemblance (I will come back to this issue later on). Likewise, this also implies that the concept of mathematical distance becomes a category of analysis and interpretation since in vector space, as we have seen, distance is not an empty notion but a measure of the degree of similarity between cultural elements (in the form of vectors). In summary, a process of cultural semantization of the concept of mathematical distance takes place; cosine similarity (the algebraic calculus) becomes an interpretative tool; and the result, namely, the similarity between cultural elements, is redefined as a measure of the degree of distance, mathematically computed, in the vector space.

The most obvious consequence of this semantization process is the adoption of a quantitatively based concept of similarity as a parameter of cultural ordering. Although this mathematical similarity intuitively works very well since, once projected onto the two-dimensional space, we can directly perceive the «resemblance» between the neighboring cultural objects under examination (see Figure 7), this must also be (and is being) discussed in terms of its divergence from the forms of similarity that are proper to human cognition and its possible reductionisms (see, for example, [22]).

At the same time, however, we can also appropriate this concept of spatial similarity based on mathematically computed distances to explore innovative ways of relating to cultural objects (which is not quantitatively based). This was one of the goals of the immersive project *Poscatálogo* (2020) developed by iArtHis_Lab [23–25]. Taking the concrete materialization of a high-dimensional vector space generated from the processing of a corpus of images with an Inception CNN as a base, we set out to explore how a notion as counterintuitive as that of high-dimensional vector space could be made to be physically experienced. For that, the two-dimensional manifold resulting from computational processing was transformed into an immersive three-dimensional space (Figure 8).

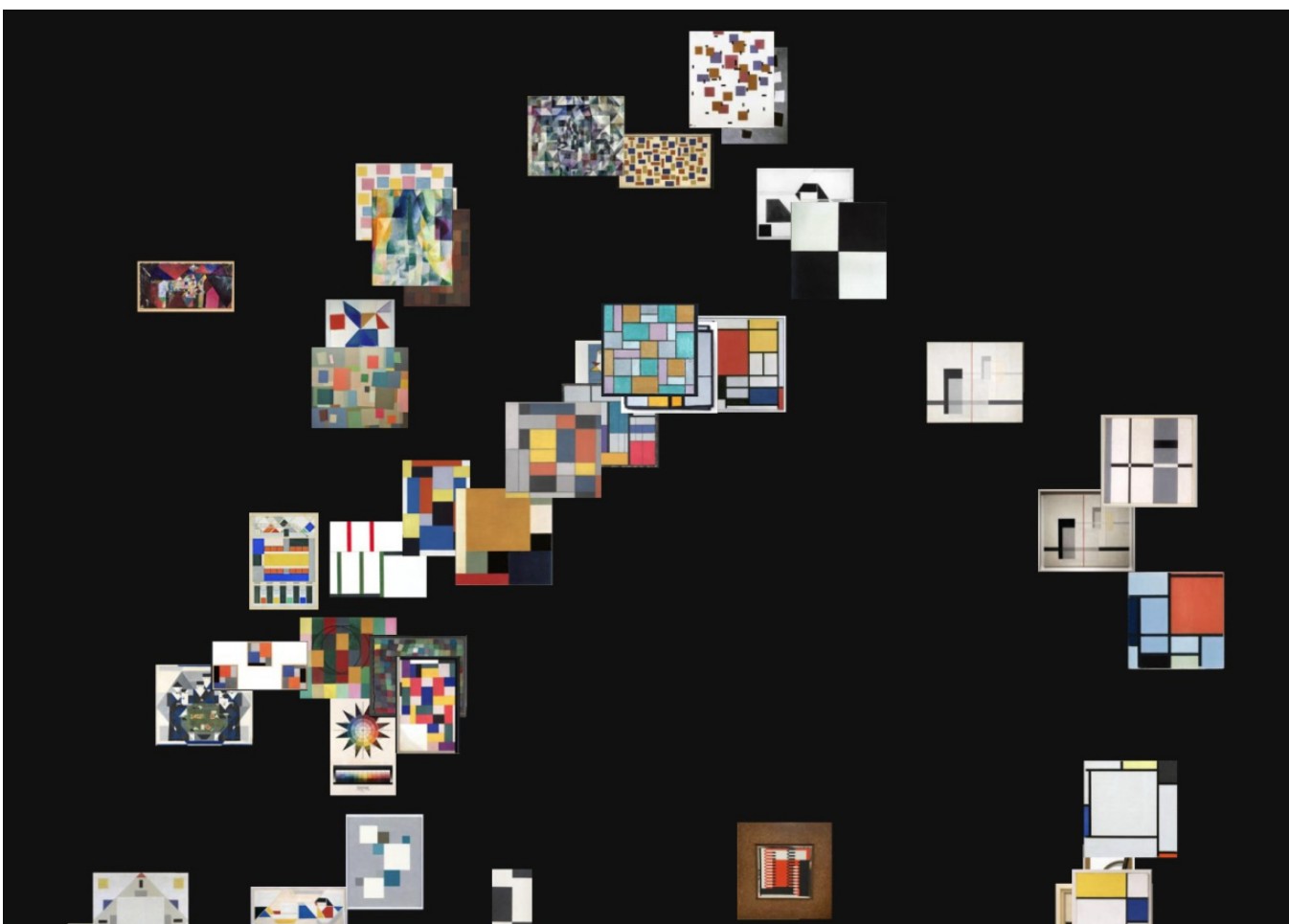

**Figure 7.** Section of a two-dimensional projection of an image vector space. Barr X Inception CNN Project. Available on: https://digital-narratives.versae.es/# (accessed on 28 July 2022) (screenshot).

In this way, the vector space, reconverted into an inhabitable, walkable environment in which the mathematical distances between the images are translated into physical distances that can be physically traversed, makes it possible to become aware of the existing gradational visual relationship between the images and their degree of similarity as the subject walks through the space; put another way, a spatial exploration of visual similarity is produced, not quantitatively but subjectively, as it is experienced with the whole body. Likewise, the bodily exploration of similarity through this performative exercise based on the traversing of physical distances between images materializes forms that are alternative to oculocentrism, that is, to the prevalence of the eye in our relationship with cultural objects in Western culture. Ultimately, my point is that while this geometric–mathematical-based epistemology can shape cultural interpretations and while, for that reason, it must be thoroughly discussed, it might or should also be the object of appropriation and reinterpretation in itself so that new (or renewed) cultural concepts can be explored. I believe that in this back-and-forth path lies the epistemological and critical broadening that these techno-concepts and techno-objects can offer us.

*Spatial Forms and Structures/Morphology*. As has been said (and can be clearly observed in Figures 4 and 5), in the analyses carried out in high-dimensional vector spaces, the visual, linguistic, textual, etc., production becomes a problem of spatial distribution, which is the factor that determines the definition of cultural objects (for example, their greater or lesser degree of similarity respect to other cultural elements) and the cultural domain in itself; meanwhile, the space acquires a certain form and structure from the positions and distances that are established between the vectors that represents the cultural elements.

Thus, the concepts of spatial structure and form become central categories for cultural analysis and interpretation.

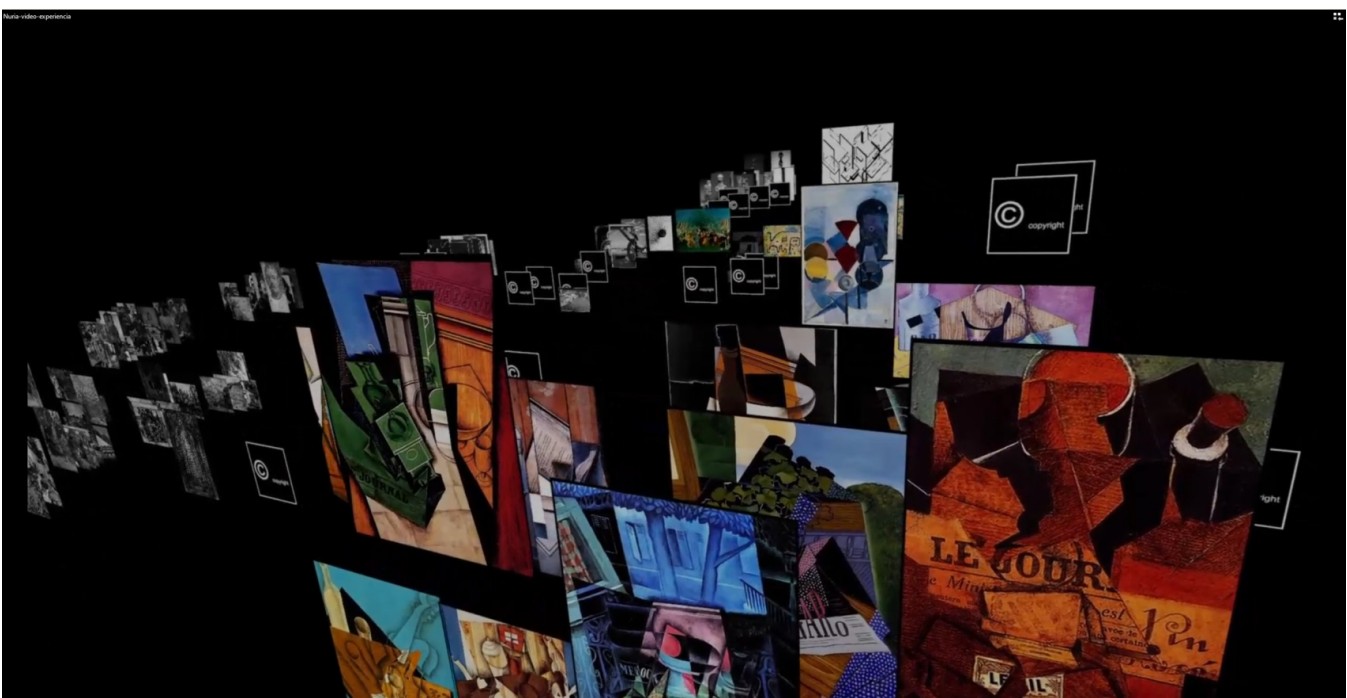

**Figure 8.** Image high-dimensional vector space rebuilt as immersive three-dimensional space. Post-catálogo Project. Available on: https://poscatalogo.iarthislab.eu/ (accessed on 28 July 2022) (screenshot).

Concerning the exploration of the structures underling high-dimensional vector spaces, topological data analysis (TDA) is a newly emerging domain comprising topology-based techniques to infer relevant features that have proved to be effective in supporting, enhancing, and augmenting both classical machine learning and deep learning models. TDA aims at providing well-founded mathematical, statistical and algorithmic methods to infer, analyze and exploit the complex topological and geometric structures underlying data that are often represented as point clouds in metric spaces [26,27]. The examination of the implications that this approach may have for cultural research and interpretation is still at a very early stage, although everything seems to point to the fact that these computational methods could provide interesting contributions. In addition to that, it cannot be disregarded that the application of these sorts of techniques oriented to explore the structures underlying data also involves the epistemological assumption that every phenomenon can be reconfigured as a structure and can be explained and interpreted in terms of structure. How this new structuralism will inform further cultural interpretations should also be an issue under discussion.

Nevertheless, my focus in this paper is exclusively on the visual configurations in which high-dimensional vector spaces materialize. In this regard, it is important to differentiate between the structure of high-dimensional space, which we do not see, and its materialization in a given visual form, which is the one we see, and, for this very reason, is the one that most strongly shapes our topo-morphological imagination. As these visual forms become discovery spaces and object knowledge in themselves, it is, therefore, essential to be aware of the distortions and deviations that these visualizations imply regarding the «real» structure of vector space; their inherent biases; the pareidolic phenomena that can occur when we try to unravel their meaning; and the epistemological assumptions that, embedded in them, endorse certain interpretations [17,28,29]. Consequently, these visualizations constitute in themselves a space for problematization, and as shapers of imaginaries and activators of certain forms of cultural understanding, they must be sys-

tematically incorporated into critical studies on contemporary visual culture. However, and without undermining this question, what interests me in this paper is their potential epistemological productivity.

These visual configurations have been the subject of different conceptualizations. Thus, they have been interpreted as a technological version of Tom Mitchell's concept of the metapicture (*Picture Theory*, 1994), as they constitute images that «talk» to us about other images, an image-within-an-image that produces a metavisual visualization [17,30,31]. Maria Giulia Dondero [31] has also evoked the concept of the self-aware image and the meta-painting proposed by Victor Stoichita for early modern paintings (*The self-aware image: An insight into early modern meta-painting*, 1997). Elsewhere [14,32], I defined these visualizations as a human–machine interface, i.e., as the intermediate space in which algorithmic logic becomes partially visible to the human subject that endows it with meaning through an interpretative process. From this point of view, it is not so much the direct correspondence that may exist between what we see and the real structure of the vector space that is decisive but its capacity to expand the epistemological imagination, raise new questions or activate intuitions toward new hypotheses, in other words, its heuristic rather than its hermeneutic capacity.

Semantic vector space models offer some examples that may adequately illustrate this argument. As their name suggests, semantic vector spaces are those that, through certain vectorization technologies, capture and encode the semantic relationships between words. The semantic vector space model is based in Zelling Harris and John Firht's distributional hypothesis, according to which words, signs, syntagms, etc., expressing similar meaning tend to be used in similar environments. Consequently, the meaning of signs can be induced through the combinatorial patterns of their co-appearances in a corpus. According to the operating logic of vector space models, t is assumed that the words that are spatially closer are those that maintain stronger semantic relationships with each other and, thus, are configured by their proximity in conceptual or semantic regions. In this sense, concepts (as semantic regions) can be understood as structures (or formations defined by a certain morphology) that result from the paradigmatic relations between words that co-appear with a given frequency (Figure 9).

Consider another example taken from LSA (latent semantic analysis), which is one of the foundational techniques of what is popularly known as topic modeling. Topic modeling is used for detecting latent topics in sets of documents by adopting, as a criterion, the frequent co-occurrence of words in the same context. In Figure 10, the uncomputed matrix does not seem to have an apparent meaning; when it is computed, however, we observe that this matrix is reorganized, acquiring a certain structure that is modeled by aggregational forms that correspond to the frequent co-appearances of words. Therefore, strictly speaking, it can be said that in this visualization, latent topics themselves manifest as forms or spatial structures.

The piece *Multiplicity* (2017) (Figure 11), which is an interactive installation devised and designed by Moritz Stefaner on the occasion of the exhibition *123 data* (2017), may also be illustrative. This installation actually constitutes a collective photographic portrait of the city of Paris constructed from hundreds of thousands of photographs that were shared on social media and were computed using a CNN and clustering algorithm that spatially distributed and grouped them according to their mathematically computed visual similarity [33]. These clusters could then be considered as latent visual themes that are embedded in the social photographic production of the city of Paris; these are clusters that, as a whole, gave a certain shape to this particular iconosphere.

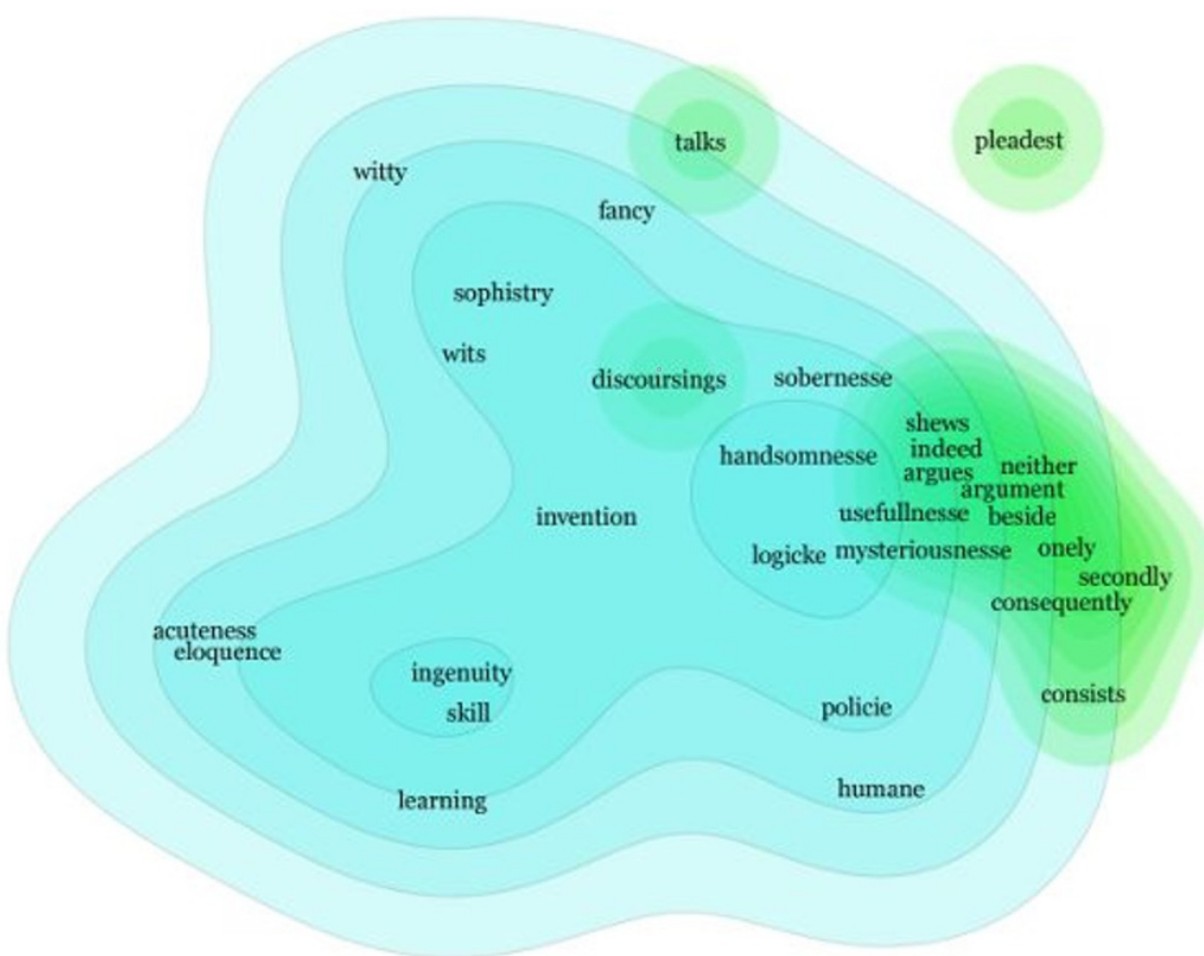

**Figure 9.** Semantic neighborhood of *wit*. M. Gavin et al. Spaces of Meaning: Conceptual History, Vector Semantics, and Close Reading. In *Debates in Digital Humanities*; Gold, M., Klein, L. F.; University of Minnesota Press: Minneapolis, 2019.

Without undermining the fact that the epistemological logic, in all cases, is the distribution of patterns and that these visual «formations» are mediated by the algorithm that produces them, these topo-morphological configurations that are now the concepts (linguistic or visual) allow us to study them as morphologies that emerge at a given moment. We can thus identify where and when they appear and how they are transformed over time by attending to how they change morphologically or how the shape of the vector space is modified owing to their trajectories and changes in position. Moreover, and more interestingly for the point of view of my argument, this exploratory context invites us to wonder about the possibility (and convenience) of developing a type of morphological semantic (linguistic and visual). For example, we could explore the association of certain topo-structures with certain types of concepts, and thus, developing a conceptual typology derived from the topological behavior of semantic domains. Given that they now have a visible form, we could also elaborate upon an iconology of concepts (visual or linguistic), and taking the comparison of their topo-structures as a basis, comparative semantics between traditionally separated domains of reality (cultural, physical, biological, etc.) could be developed, which might facilitate a more holistic understanding of the configuration of the world.

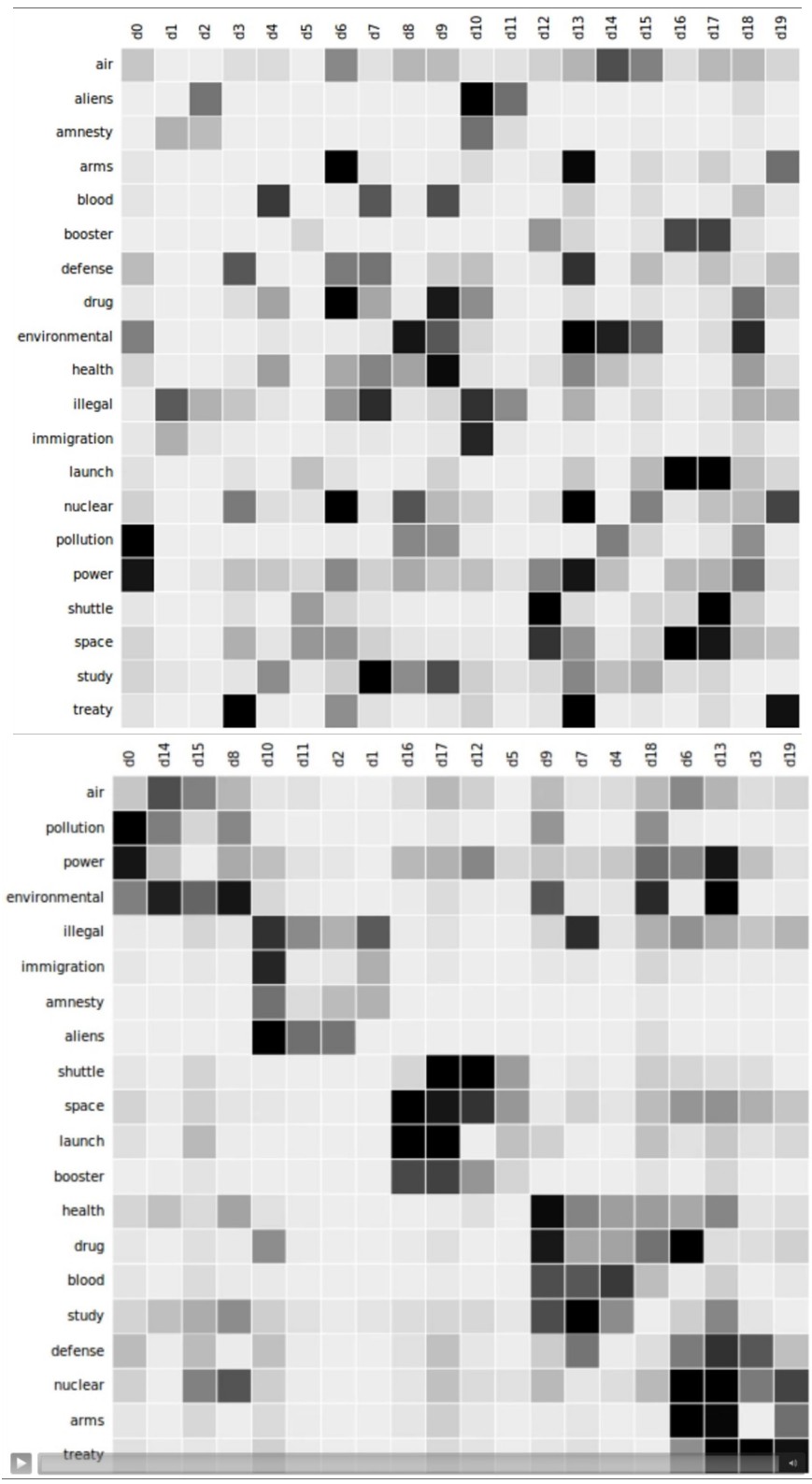

**Figure 10.** Top: data matriz uncomputed. Bottom: data matrix after being computed. Cristoph Carl Kling. *Animation of the topic detection process in a document-word matrix* (2017). Available on: https://commons.wikimedia.org/wiki/File:Topic_detection_in_a_document-word_matrix.gif (accessed on 28 July 2022) (screenshot).

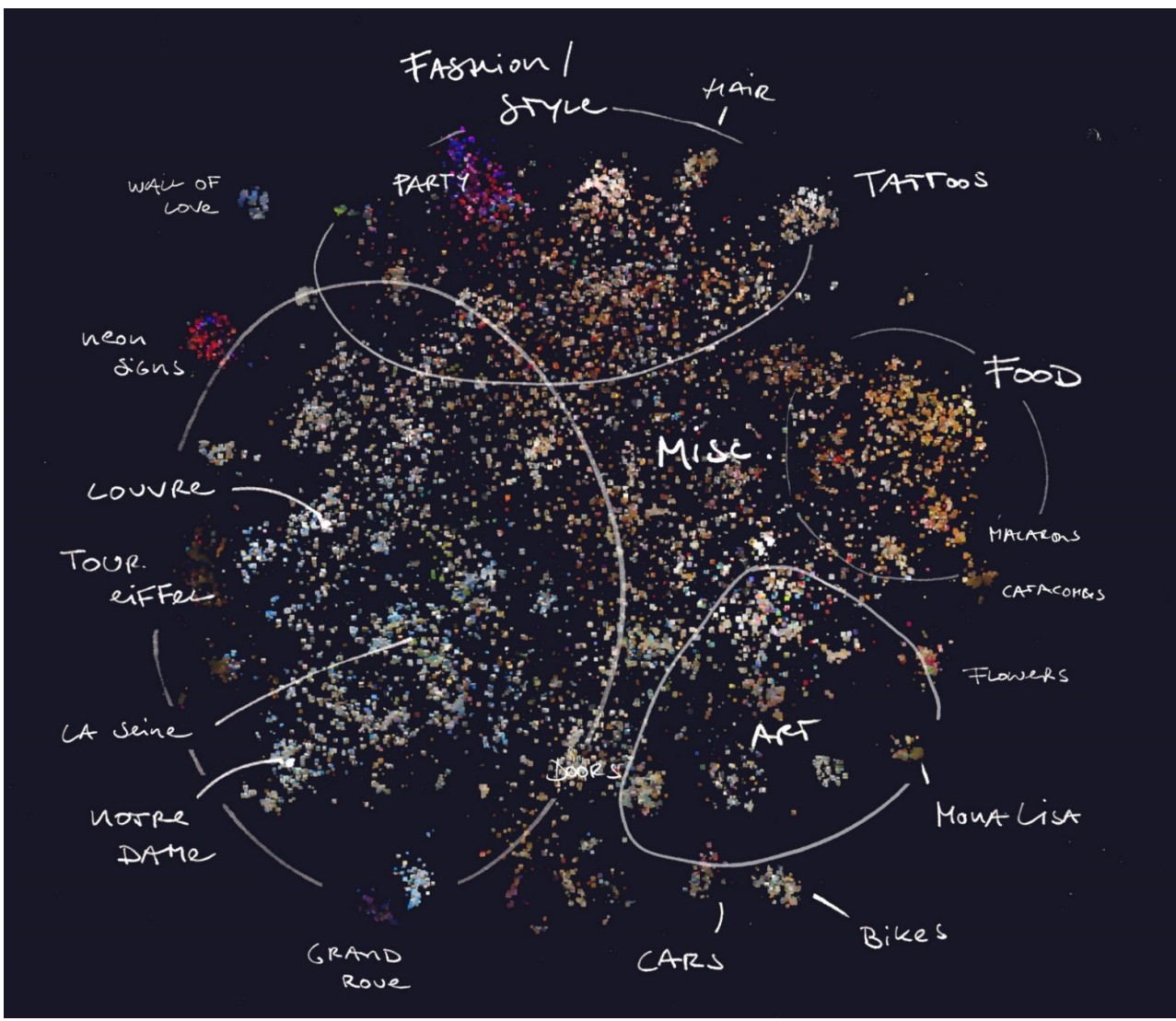

**Figure 11.** Paris' image clusters. Topic's clusters are manually identified by the artist. Moritz Stefaner. *Multiplicity* (*1 2 3 data* exhibition, 2017). Available on: https://truth-and-beauty.net/projects/multiplicity (accessed on 28 July 2022).

It should be stated that this proposal is far from new. In this regard, it is worth remembering that in his 1981 article *Spatial Form in Literature: Toward a General Theory*, Tom Mitchell reiterated that «the concept of spatial form has unquestionably been central to modern criticism [ ... ] Indeed, the consistent goal of the natural and human sciences in the twentieth century has been the discovery and/or construction of synchronic structural models to account for concrete phenomena» [34] (p. 539). Following this reminder, Mitchell delved into what a general theory of literature based on a spatial form could consist of. Just one year later, he proposed the development of a diagrammatology, which was defined as the systematic and historical study of the operative forms in the literary tradition [35]. Likewise, Maria Giulia Dondero, within the framework of her semiotic approach to these techno-images, proposed the concept of diagramma of images (or diagrammatic images), taking as a basis the notion of the diagram in the perspectives of Charles Sanders Peirce, Gilles Deleuze and Nelson Goodman [36].

In summary, these techno-images (or techno-forms), as boundary entities between the computational rationale and the human thought-/imagination, the visible and the invisible, the mathematical structure and the visual form, constitute an interesting space for interdisciplinary exploration in which machine learning, visual studies, visual semiotics

and morphology should converge. Likewise, they constitute an interesting context in which to explore the epistemological potentialities of previous theoretical frameworks for the configuration of a possible topo-morphological epistemology.

### 3.3. Topological Narratives

The spatialization of the forms of reasoning that are inherent to high-dimensional vector spaces, their algorithmic logic based on the detection of patterns (and the distribution of vectors according to their similarity with the pattern encoded in a model) and their materialization in topo-morphological images lead to topological modalities of narratives, where concepts such as connectivity, relationality, continuity, proximity gradients and transformation (among others) prevail. All of these are concepts that are associated with topology, which are adopted (explicitly or implicitly) as categories of analysis and interpretation when dealing with high-dimensional vector spaces. As indicated in previous paragraphs, topological thinking is not new in the field of art historical and/or cultural interpretation; however, the growing «topologization» of discourses and narratives around cultural heritage associated with the phenomenon of cultural vectorization and its computational processes demands that it be given a new centrality in the theoretical–critical discussion, taking into account, moreover, that our horizon of reflection is now a techno-topology. In this regard, it is important to bear two issues in mind. On the one hand, it is necessary to explore the epistemological continuities and discontinuities regarding previous topological approaches in the field of art history and cultural heritage. For example, the visual similarity between Warburg's *tafeln* and the visual feature spaces in which high-dimensional vector spaces materialize, and the fact that neural networks used in computer vision are capable to detect invariant patrons over time, cannot make us lose sight of the fact that Warburg's topological thinking about the persistence and transformation of images is completely different from the algorithmic logic of deep learning technologies, no matter how much we try to interpret them from the Warburgian perspective. On the other hand, it is necessary to detect the problematization fields to which these topological narratives give rise. Let us consider two examples.

*Neighborhood/Proximity/Affinity/Similarity*. As has been repeated throughout this paper, the logic underlying high-dimensional vector spaces makes it possible to materialize the concept of neighborhood (semantic, visual, linguistic), which tends toward a study of cultural phenomena in terms of contiguity and affinity. The reorganization of visual or linguistic production, based on neighborhoods of related elements, undoubtedly contributes to a diverse understanding of cultural production by emphasizing contiguity beyond traditional classificatory schemes, which enables interconnections between cultural objects that are sometimes located in very different places in those epistemological systems built according to traditional categories of ordering and classification. These cultural reorganizations also provide interesting materials to propose alternative narratives to the traditional chronotope regime (geospatial and linear–chronological) by allowing us to inquire into nonlinear temporalities and spatial relations differently from those that are geographical or geopolitical. In this sense, the concept of contiguity, which is associated with the idea of neighborhood, also encourages the development of horizontal narratives, which favors a very productive context for experimenting with transchronological, de-hierarchical and nonstemmatic (or nongenealogical) narratives. The forms of contiguity and proximity also support, as Remedios Zafra claims [37], thinking in terms of affinity and gradient as opposed to the «identitarian, dual and excluding forms» (p. 121) that are characteristic of those classification systems that divide or separate. For Zafra, the phenomena of erosion, fusion and confluence constitute the distinctive forms of our fluid and networked culture, which demands other forms of thought that make integration and conciliation possible.

However, we must not forget that these narratives of affinity and contiguity are, in reality, narratives of similarity and resemblance since, as we have seen, the proximities in a vector space are nothing other than degrees of similarity. As we know, the concept of similarity is plural and the analysis of the similarity that relate cultural objects, processes

and phenomena to each other can respond to different dimensions depending on the research interests. However, the algorithmic modeling of similarity imposes the same logic in all cases: mathematically measured proximity to a pattern or model. Instead, human similarity does not operate only from the recognition of patterns, it also works by evocation and resonance, it is culturally modeled, and it puts into play other complex forms of similarity, such as metaphor.

In addition to delve into what similarity means in algorithmic terms, it is also necessary to discuss what it means to rely on narratives that are based primarily on similarity and resemblance. As José Luis Brea pointed out some years ago [38] (pp. 82–83), AI imposes a logic of resemblance or, in other words, of recognition through mere resemblance. Rather than being different, cultural objects (transformed into vectors) «seems» to be more or less similar in the high-dimensional vector space. In this respect, I think it could be interesting to establish a distinction between the notions of dissimilarity and difference. Dissimilarity can be qualified according to the degree of resemblance between the things under analysis (objects can be more or less dissimilar); difference, instead, according to their specificity and singularity. This circumstance, together with the fact that models produced by the AI technologies are the materialization of invariant feature structures extracted from heterogeneous datasets, raises the crucial problem of how to approach the difference. These models are extremely valuable for analyzing the not-immediately-evident communality underlying cultural productions and processes. This potential explains their increasing use to extract the set of shared, recurrent characteristics that shape cultural and visual production but, at the same time, complicate the recognition of diversity, the new, the unique, the uncommon and the disruptive [9]. This is why, together with the algorithmicity of the concept of similarity (and its potential reductionism), the relationship between similarity and difference, as well as the notion of difference in a vectorized culture, constitutes one of the epistemological problems that must be addressed.

*Continuity/Transformation/Transition/Transductivity*. The topological narratives underpinned by high-dimensional vector spaces also advocate an approach to the study and interpretation of culture in terms of continuity or, more specifically, of continuity «in» transformation. It must be kept in mind that in the high-dimensional vector space, cultural objects are no longer perfectly delimited entities, with each located in the box that corresponds to it according to its inherent, fixed and stable properties (as in the grid model at the basis of the traditional concept of archive); rather, they become a set of (numerical) characteristics that represent a point in a space that is made up of continuous dimensions. There (not as an object, but as such a set of characteristics) they maintain relations of degree with the rest of the points. That is why high dimensional vector spaces facilitate the exploration of the fuzzy boundaries that exist between cultural elements (words, images, concepts, etc.) that are not breaks but are more or less continuous transitions. These topological narratives thus constitute a suggestive (although not new) alternative to narrative modalities based on polarization, delimitation and bounding.

Figure 12, for example, shows us the evolution of Western pictorial production as a single topological form made up of regions and intermediate spaces that operate as continuous transitions between them. The image of art history as a sequential evolution of periods, styles, isms, poetics, etc., vanishes to become a unique space of characteristics, which has been explored unevenly over time by artistic practices but without discontinuity or rupture. Moreover, this vision of the totality in a synchronic manner is not timeless but spatiotemporal since time (inscribed in the process of transformation) is spatially modulated, or it can also be said that space is modulated in the temporality of what is transformed. Thus, the possibility of developing more complex temporalities that escape chronological regulation also emerge in the context of high-dimensional vector spaces.

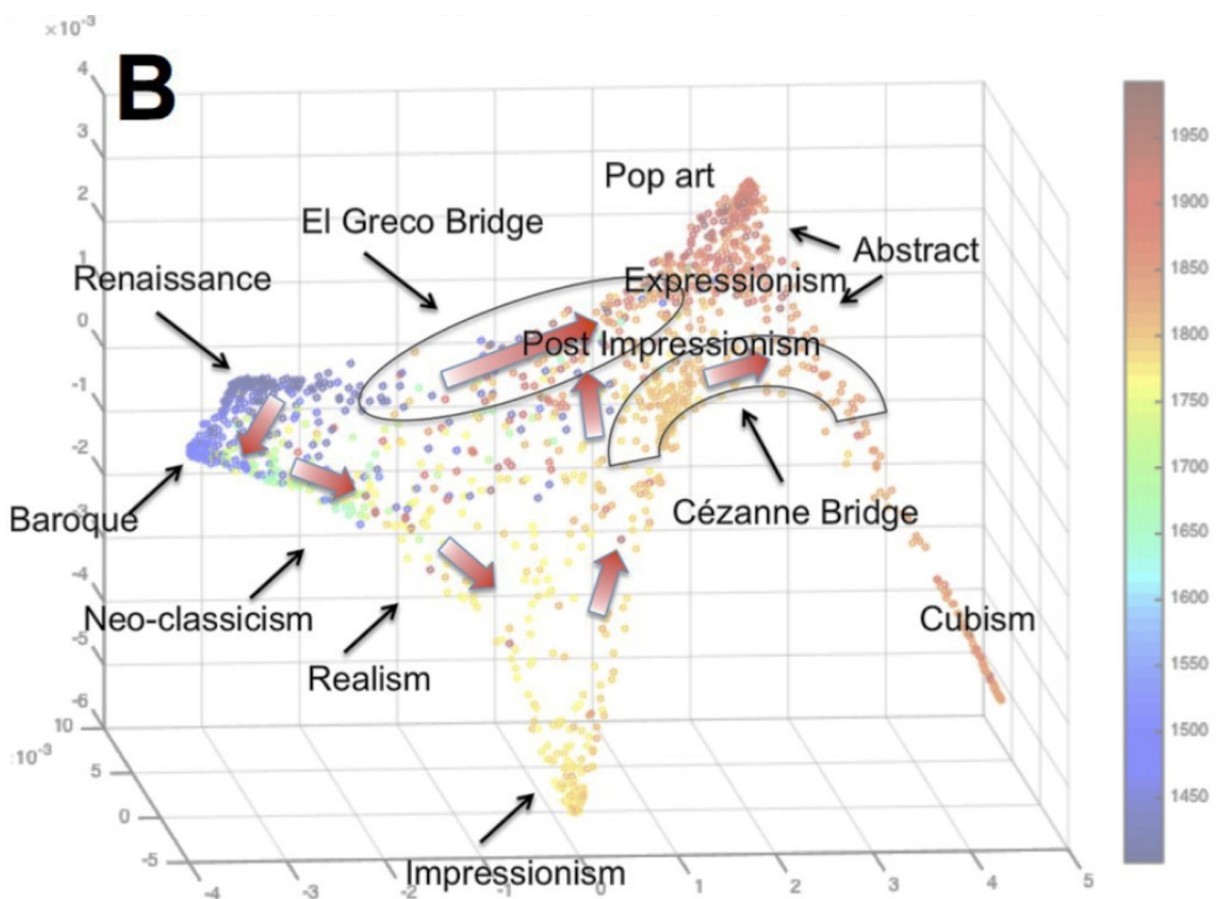

**Figure 12.** Evolution of Western pictorial production projected in a three−dimensional manifold. A. Elgammal et al. The shape of Art History in the Eyes of the Machine. *arXiv: 1801.0772* **2018** (Figure 8B).

Nevertheless, and regardless of the research potentialities involved in these approaches, the question about the extent to which introducing continuity into the discrete entails heuristic values and to what extent this leads us an unfertile distortion cannot be obliterated. Likewise, thinking in terms of continuity also involves bringing to the center of the debate its counterpart, discontinuity. Therefore, it is the dialectic between continuity and discontinuity as behavioral logics of cultural phenomena that truly becomes epistemologically productive.

If we take into account the fact that latent vector spaces are, in reality, regions of encoded features, it is easy to infer that the movement through a latent vector space entails a progressive transformation of the observed characteristics. If we examine it, for example, in the realm of images, the iconosphere is presented to us as a visual continuum that transforms in a progressive and nonlinear way, as we observe in Figure 13, where images progressively transform from left to right. This transformative imagery raises a new order of questions; for example, what properties or characteristics could be considered critical to determining that a visual form is constituted as a distinct entity? What is the threshold that leads us to recognize forms as distinct entities?

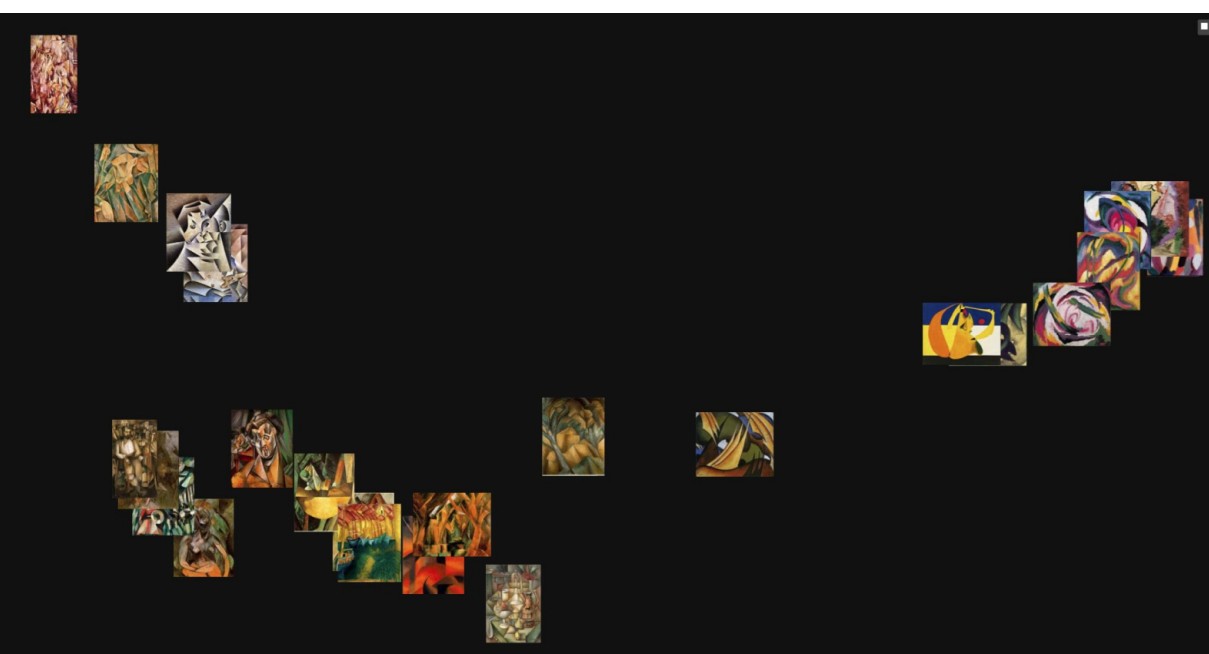

**Figure 13.** Section of a two−dimensional projection of image vector space (Barr X Inception CNN). Available on: https://digital-narratives.versae.es/#. (accessed on 28 July 2022) (screenshot).

The question of the transformation and permanence of cultural contents, objects and forms is a *topos* of cultural research. High-dimensional vector space, however, propitiates a shift from the cultural production, that is, what is subject to change, to the process of transformation itself. The problem we confront (or should confront) is not a culture that transforms, trying to elucidate how we get from one point to another, but a «culture-in-transformation», so that it is the very process of transformation that becomes the object of exploration insofar transformation (in other words, the dynamics of change) is inherent to culture. This issue is illustrated when we understand vector space as a space in which multiple directions unfold. We must remember that a vector is also a trajectory and that a vector space has not only a magnitude but also a direction. The dimensions of the vector space specify the number of directions in the space. In addition, directions in the latent space can encode specific aspects of the cultural domain being analyzed. Therefore, the notion of direction can be used as an investigative tool in itself since the exploration of directions, i.e., the path between two points in the latent space, allows us to examine the chains of characteristic variations that unfold between them (Figures 13 and 14). Thus, this quality of latent vector spaces facilitates an understanding of culture as a continuity «in» transformation, as a continual becoming, meanwhile placing the concepts of transitionality and transformativity at the center of the epistemological inquiry insofar as we can now materialize and «see», in a concrete way, the grade, scale, intensity and range of transformations operating between regions of cultural domains.

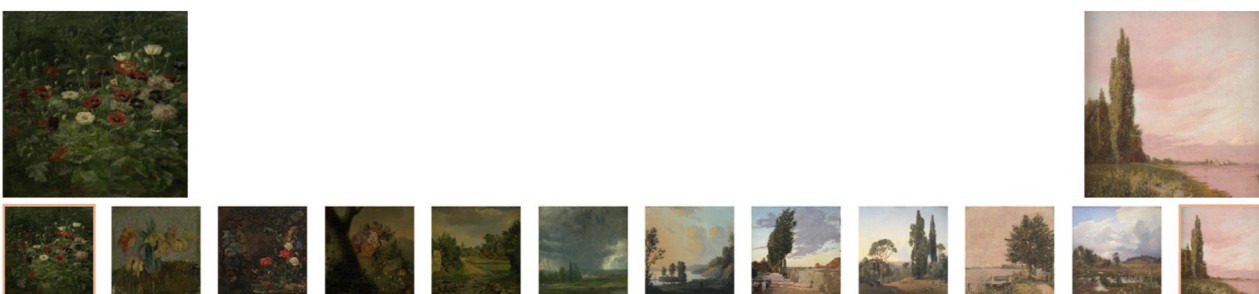

**Figure 14.** Walk through the high-dimensional space from one artwork to another. D. Bönisch. The Curator's Machine: Clustering of Museum Collection Data Through Annotation of Hidden Connection Patterns Between Artworks. *Digital Art History Journal* **2020–2021**, *5*, 21–33 (Figure 5).

The continuity we have been discussing is also manifested from another point of view: multimodal continuity. This continuity can be seen as the result of the ontological equalization of different semiotic systems that are derived from the transformation of cultural objects into numerical information entities and vectors. From a computational point of view, there is no substantive, material or ontological difference between words, images, sounds, etc.—they all have the same materiality and mode of existence. In fact, strictly speaking, it could be said that multimodality itself vanishes in the vectorial continuity. This circumstance implies that vectors of images and words (for example) can coexist in the same vector space, making it possible to compare and analyze them together or even, due to the porosity of neural networks, making possible the production of texts from a given image or the generation of synthetic images from texts. See, for example [39–41] regarding the production of captions and texts associated to artworks images. Perhaps, the most well-known models that handle with the task of image descriptions and text-to-image generation are OpenAI's CLIP, DALL-E 2 and GLIDE, which have become very popular in recent times. Likewise, the so−called multimodal AI is based on the construction of exploratory systems that combine different analysis technologies (NLP, CV, etc.) and integrate multiple semiotic modalities to produce meanings that emerge from the synthesis between language, images, videos, etc. The operating logics and the outputs generated by these models really seems to lead us to a post-visual and post-linguistic era marked by a (post-human) transductivity between semiotic systems.

Although we are still at an early stage in the field of cultural heritage, this emerging scenario prompts us to inquire about what avenues of exploration are opening up and what areas of problematization are emerging beyond the possibility of producing image captions/descriptions or synthetic images. Undoubtedly, cultural vectorialization, high-dimensional vector spaces and latent spaces offer us a very valuable framework to explore transfers and transitions between images and words in a different way than we have done so far. From my point of view, one of the most striking projects regarding this question is *The Next Biennial Should be Curated by a Machine. Experiment: A-TNB* (2021), designed and developed by Joasia Krysa, Leonardo Impett and Eva Cetinic [42]. This experimental project uses CLIP (Contrastive Language–Image Pre-training) to explore cross-similarities between images and texts, so that a sort of transductive ecosystem is created in which synthetics images generated from texts and generative texts produced from images are related according to their degree of similarity. Since the generated images are created only from titles, the most similar generated images are therefore connected through the visual similarity of their (textual) titles. In the same way, since the images' textual descriptions are generated from their visual features, the most similar texts are connected through the textual similarity of their (visual) appearance [Figures 15 and 16]. Certainly, it is only an experiment, but it is compelling enough to invite us to reflect on what other dimensions of similarity exist (or can exist) that scape human intellectual and cognitive mechanisms, as well as involving approaching visual-textual transfers and transitions on another order of complexity.

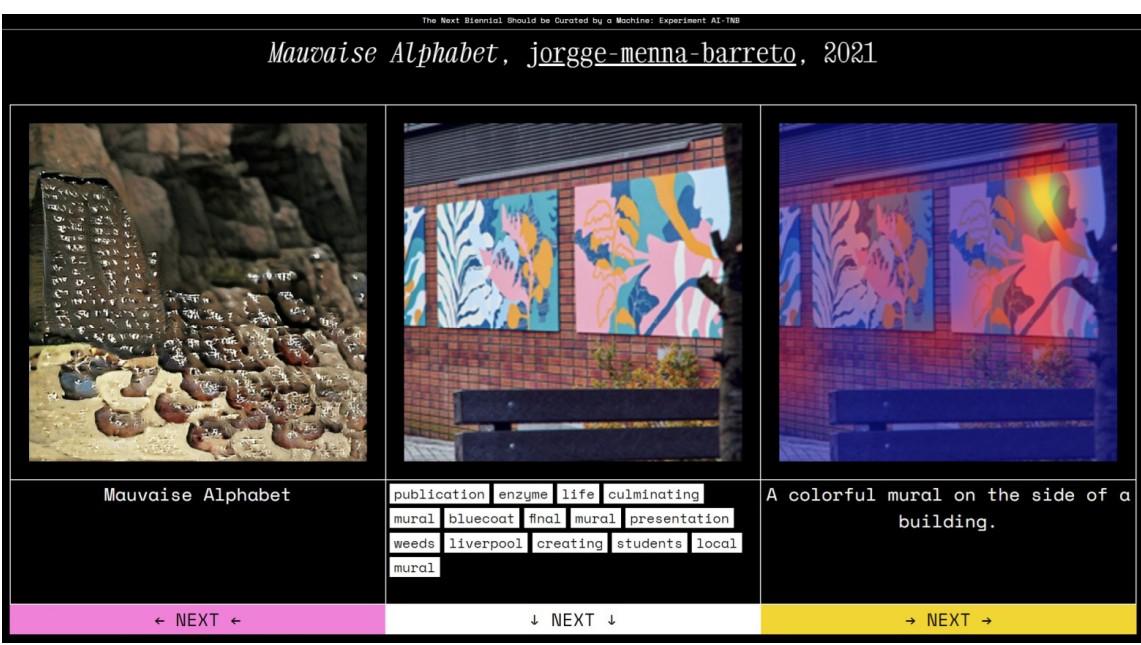

**Figure 15.** Triptych of the *The Next Biennial Should be Curated by a Machine.* On the right, a textual description generated from the visual features of the image on top is shown. Joasia Krysa, Leonardo Impett and Eva Cetinic. The Next Biennial Should Be Curated by a Machine. Experiment: AI-TNB. Available online: https://metaobjects.org/testing/liverpoolbiennial/ (accessed on 18 July 2022) (screenshot).

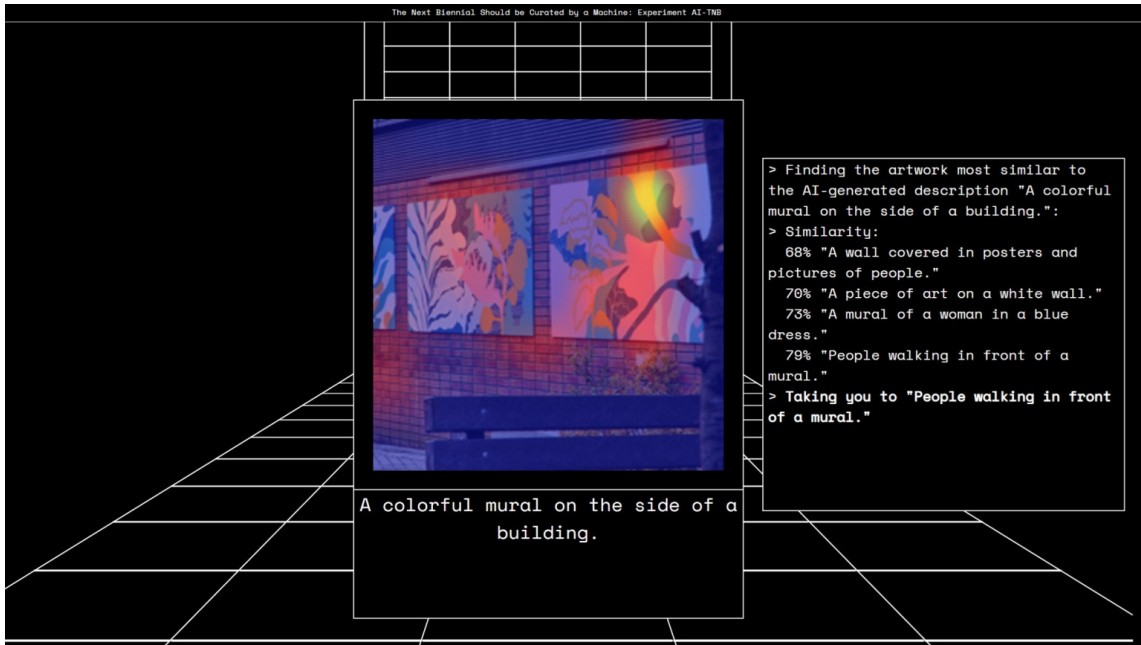

**Figure 16.** Ranking of most similar artworks on the basis of their AI-generated descriptions. Joasia Krysa, Leonardo Impett and Eva Cetinic. The Next Biennial Should Be Curated by a Machine. Experiment: AI-TNB. Available online: https://metaobjects.org/testing/liverpoolbiennial/ (accessed on 18 July 2022) (screenshot).

The fluid boundaries between words and images (or, in other words, the linguisticity of the image and the visuality of the language) is a cornerstone of Tom Mitchell's picture theory (to cite perhaps the most widely known example); therefore, here, we do not again

confront an unprecedented paradigm of cultural interpretation. However, these models provide us with concrete tools to make transductivity a central category for cultural analysis and interpretation.

In summary, beyond narrative and research potentialities, the essential question at stake is how to re-operationalize these not-new concepts (continuity, transformativity, transitionality, transductivity), but now re-modeled by the techno-topologies generated by computational methods in order to produce truly relevant knowledge in the field of cultural research and interpretation. Likewise, these topological narratives also reposition in a new context the debate around the dialectical interplay between transformations and invariants, deformations and permanence, as well as the tension between continuity and discretization. How to go beyond the already known intellectual concerns associated to these dialectical interplays is also a crucial question.

## 4. Generativity

The versatility of high-dimensional vector space models described in the previous sections also relates to their condition of productive–generative spaces, which allows us to (and demands that we do) extend the reflection beyond their representational function. In fact, one of the definitions usually given for a vector space is that of a state for a system with many degrees of freedom. As previously explained, a cultural object encoded into a vector constitute a point among multiple dimensions in the high-dimensional vector space; it is a state among multiple possibilities. The interesting issue is that AI technologies allow us to explore these other possibilities and shape them as we move through the latent vector space not to already existing points but to completely new points. In fact, the transformativity and transductivity that vector space models embody are generative in themselves since the process of transformation and transduction involves the generation of something new when exploring the latent vector space. This is why latent vector space models have been demonstrated to be extremely interesting for contemporary artistic experimentation and other tasks related to visual and linguistic production.

The generativity of high-dimensional vector spaces requires a deeper and more detailed examination; however, I would like to highlight just two issues before concluding this (already very long) paper. First, visual and linguistic generation could be conceptualized as exploratory and investigative instruments by themselves; that is, as a form of art-historical and cultural research that would no longer be based on what actually exists (what we can analyze and interpret) but on what does not exist but is possible, insofar as AI can model it and bring it into existence. This opens up the possibility of a speculative and contrafactual history of culture based on the generative quality of vector space models.

Second, through vectors and vector spaces as the new devices for recording, preserving and ordering cultural heritage (as was earlier suggested), the concept of archive could be reformulated in a generative sense. A generative archive would adopt DNA as a model. DNA preserves and maintains information from generation to generation (a memory) meanwhile generates individuals who are always new and unique through performing processes of differentiation. Therefore, when faced with the cultural model of archive memory (retentive and recuperative), could we consider a shift to the concept of genetic code as a basis for its rethinking? This concept of the archive would go beyond the idea of building new narratives, arguments or hypotheses on the basis of the materials already existing and recorded in archives. It would also not be about detecting silence and gaps in archives to propose alternative narratives. The generativity of an archive, thought as genetic code, would be about producing new cultural objects and materials, new cultural «individuals» with the capacity to expand the comprehension of cultural manifestations and processes as complex phenomena. Would it be possible? What kinds of devices and practices could we develop to give rise to this «archive» as a genetic code of open possibilities that is in continuous generation and becoming?

## 5. Final Remarks

This paper advocated an epistemological turn in the field of digital art history and cultural heritage studies. This epistemological turn is understood as the elaboration of a new (or renewed) epistemic apparatus that allows us to approach and interpret cultural phenomena from the perspective of a different order of thought. This different order of thought involves going beyond providing new answers to questions already posed in the field of art history and cultural heritage in order to ask new questions, since only when questions are changed (and not only the answers) can it be affirmed that an intellectual and epistemological paradigm shift occurs. Therefore, this epistemological turn should also entail an opening towards intellectual concerns different from those that have traditionally been part of art history and cultural heritage studies. This epistemological labor has been defined as an «epistemological technical practice», which means thinking, epistemological reflection and production and the technical making and design as mutually imbricated in an undividable process. Furthermore, I consider that the differential factor of digital humanities with respect to other humanities and with respect to other theoretical approaches to techno-cultural issues should reside precisely in the fact that the digital humanities have the capacity to engender thoughts and arguments through technology, and not only «about» technology.

The imbrication between techno-scientific making and intellectual production is also at the basis of the idea of a techno-concept, which was proposed in this paper as a notion that could contribute to shaping the advocated epistemological turn. This paper presented the techno-concept as an abstract category that refers to those intellectual productions in the form of ideas and/or concepts, whose elaboration is mediated (influenced, determined, conditioned) by the same technology that is used for their operationalization. In other words, such ideas and concepts, in their current formulation, would not exist as such without the technological mediation. That is the reason for which this paper argued that those concepts/ideas should be understood as co-productions between human and techno-computational instances. Subsequently, techno-concept is also a notion that reinforces the need to approach epistemology from a posthuman perspective in order to overcome anthropocentric reductionisms.

As particular cases, this paper argued that the mathematical concepts of n-dimensional, vector and latent spaces constitute examples of techno-concepts that can be reappropriated and reworked for cultural analysis and interpretation. The centrality currently acquired by these concepts as a consequence of the increasing transformation of cultural objects and phenomena into vectors within the framework of ML and deep learning applications explained the interest in focusing the attention on those concepts re-thoughts as techno-concepts. This paper provided a preliminary inquiry, in which only a small number of epistemological implications and derivations have been exposed, although it has made it possible to unfold some interesting issues. Thus, the inquiry on the concepts of n-dimensional, vector and latent spaces made visible a network or constellation of interconnected concepts inscribed in the technological practices associated to these mathematical spaces. This set of concepts (neighborhood, contiguity, affinity, continuity, transformativity, transitionality, transductivity, generativity, etc.), which are also expressive of the condition that define our post-digital society (hyperconnected, fluid, entangled, processual, datafied and modeled by an increasing-pervasive AI), configure an embryonic array of concepts with which to elaborate a remodeled epistemological body for cultural analysis and interpretations. It was also expounded how these concepts shape certain narrative modalities that, likewise, must be discussed on the basis of the epistemological assumptions embedded in the algorithmic logics that sustain them.

These techno-concepts, together with the techno-objects (or technological devices produced by the computational methods in themselves that have the capacity of modeling our imagination), bring to the center of the debate concepts that, although not new for cultural analysis and interpretation, require being reworked within the techno-mediated framework in which they are now inscribed, while they also demand research to investigate how they

can be operationalized for the production of truly meaningful and non-banal knowledge. This epistemological operationalization must be accompanied by a critical discussion of their conceptual limits or possible distortions, but also by innovative and creative reappropriations as means to increase their epistemological fertility. In this regard, heuristic approaches (and not only hermeneutic) can play an interesting role, as much as epistemological elaborations and creative practices should be conceived entangled processes.

Complementarily, a historical study must also be part of the programmatic inquiry into a techno-concept to allow it to be placed in continuity with previous formulations or types of proto-techno-concepts so that its formation process and lineage, the continuities and discontinuities that emerge through time and space can be adequately understood. Rooting these techno-concepts in previous formulations and theoretical frameworks is also a way of redirecting research in the field of digital art history and cultural heritage to explore not only how techno-scientific advancements can expand these fields, but also how their intellectual and epistemological traditions can enrich and reformulate techno-scientific developments.

**Funding:** This research was funded by the Ministerio de Ciencia e Innovación (Spanish Government), grant number PID2021-125037NB-I00, and by the Junta de Andalucía, grant numbers PY20_00508, UMA20-FEDERJA-126.

**Institutional Review Board Statement:** Not applicable.

**Informed Consent Statement:** Not applicable.

**Data Availability Statement:** Data is contained within the article.

**Acknowledgments:** I would like to thank reviewers for the detailed revision and their invaluable comments and suggestions. My special thanks to Leonardo Impett and M.ª Luisa Díez Platas for their help with the mathematical concepts.

**Conflicts of Interest:** The author declares no conflict of interest.

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
