# Peer review of "Techno-Concepts for the Cultural Field: n-Dimensional Space and Its Conceptual Constellation"

_mti, doi:10.3390/mti6110096_

Round 1

Reviewer 1 Report

This paper aims to change how cultural analysis and interpretation are codified in art history and heritage studies by introducing a new digital humanities paradigm based on the mathematical concepts of n-dimensional, vector, and latent spaces.

This proposition is quite interesting and, in my opinion, innovative, especially because of its strong interdisciplinary value across computational and humanistic disciplines.

Your critique of AI (and Machine Learning) and utilizing patterns to decode culture is relevant. It delves deep into how these techniques reshape our epistemic capabilities and explains their limits.

The article is well written (kudos). However, it is very long (you also self-critically recognized this on page 27). I feel that some sections must be fixed and shortened for clarity and conciseness; the overall structure can be improved as your very long sections make it hard for the reader to follow your thoughts. Most importantly, you failed to provide proper citations in some sections (especially section 1 – see my comments below).

Here are some suggestions for improvements:

1. Techno-Concepts for the Cultural Field

In this intro section, the author is upfront about her arguments. Clearly, it defines the theoretical framework that allows her to develop them through the definition of the epistemological value of techno-concepts.

However, the section is quite long (3 pages and ½). It must be shortened or split into multiple sections (e.g., Introduction (with aims and arguments) + Techno-Concepts for the Cultural Field (where you can provide readers with details on the topic). The proper introduction starts at line 54 on page 2 and lacks clarity as it seems to go on for too long and introduces too many aims and goals (the latest even on page 4 of the paper).

I suggest using more references and avoiding lengthy background information or explanations that may turn off your readers.

Line 27 mentions “many of these projects” but does not include references.

Line 106. It is unclear if “n-Dimensional Space and its Associated Concepts” is a subheading/subsection. I am unfamiliar with the journal guidelines for authors, but it would be best to add numbering or reformat (e.g., 1.1 – I see the same issue in the following sections). Also, this is the core of section 1 but comes after over two pages of other info.

Line 115, here again, you mentioned previous studies but did not cite any. Your reader may want to access these references you are mentioning. Correct this recurring issue; otherwise, your introduction section is incomplete.

Line 143-146. It is a matter of style, but you use one too many transitions between sentences (i.e.,” Therefore, and in accordance with the objectives of this paper, rather 143

2 Regarding these concepts, the contributions by Pasquinelli and Joler [9], Offert [10, 11], and Offers and Bell [12] deserve to be highlighted.

Multimodal Technol. Interact. 2022, 6, x FOR PEER REVIEW 4 of 31

 than examining these analytical possibilities in the following sections” or “Furthermore, it should be kept in mind 146 that, as we will see below”) that are more appropriate for a book or long essay than a journal article. I would suggest refraining from using so many transitions for clarity purposes.

2. n-Dimensional, Vector, and Latent Spaces

This section is quite long and could be shortened. I would recommend using subheadings to break the section into more focused subsections.

Lin 184: no need to include this “(1024 = 32 × 32)

Lines 226-228: This statement is imprecise or wrong. Not all AI is ML. And what you seem to be referring to is a specific technique within ML called transfer learning.

289-292 Again, the following statement is imprecise/false: “Furthermore, it must be considered that AI can 289 only operate within the framework constituted by the training dataset.” Be clear when referring to ML and when to AI.

The figures in this section are good and provide readers with a visual way to access complex concepts. It is unclear whether you hold the copyright or made them yourself (this may be discipline-specific).  

I would suggest improving the captions using clear language such as ‘Graph 1 Source: Artist A + URL’; ‘Figure 2 Source; author X + reference’; ‘Map 3 Source: your name; etc.’). The language you used in the caption is inconsistent (i.e., sometimes you used “Taken from:” other times, you just mentioned the author's name. Use conventions from your field but be consistent.

Discussion of the Postcatálogo Project is relevant and advances your argument well.

3. Problematization Fields and Epistemological Implications

This section is very good as it provides many examples that help the reader understand your proposed framework and support your argument. Leave it as it is and address my comment on improving figure captions.

Final remarks:

This section must be improved as it includes weak language and prepositions that undermine your long lines of thought and argumentation on the topic.

Lines 856-859, I would avoid repeating that “this proposal will have to be discussed and revised in further studies through dialogue with the research community” - it makes your argument weaker.

Since this section is your Conclusion, I would use the past tense when you repeat here the goals of your paper or your argument. E.g., this paper aimed at fostering discussion on topic X… and did so and so to achieve that. This paper argued that …

Lines 867-859: again, here, I would avoid justifying what you did not do (e.g., using language such as this is “very preliminary” and this is “non-exhaustive.” Your paper is mostly theoretical, so you do not have conclusive results is fine with me.

Reviewer 2 Report

The characterization of artifacts borrowed from cultural heritage or digital humanities as an object in an n-dimensional space is not new per se since this exists already for many years, especially since the advent of sensors (which return dataflows in those terms). Furthermore, this characterization is incomplete: artifacts can be scalar, vector, bi-vector, multivector, tensor, etc. Not just a vector in a space of n dimensions.

Second, there is an ontological confusion in this paper between the dimension of such a space and the length of a dimension. P. 4 reports that the image in Fig. 1 is of 1024 dimensions. This is incorrect. It is a 2x2 matrix of scalars. Therefore, the entire artifact consists of only three dimensions, not 1024. However, the lengths of these dimensions are 32, 32, and 1. When this image is characterized by more variables, it will be a 2x2 matrix of vectors. For the same reasons, the example in fig. 2 is not a 27-dimensional space but a simple 3D space. Section 3.2 describes this space as a spatial, geometric, and topological space. It is not always topological, but it can be always algebraic.

The so-called techno-concept introduced in the paper is actually a conceptual model of the artifact belonging to the referred domain. So, per se, this is not really new either. It does not offer any further characterization.

"Specific dimensionality reduction algorithms" have been primarily introduced not for facilitating the understanding of the domain or the manipulation of artifacts from this domain, but mainly to reduce the computational complexity of the problem and become more efficient.

Some sentences are sometimes difficult to understand. For example, what is a "statistical hallucination"?

Some other sentences need mitigation. For example, "nor can it generate anything new beyond it" is no longer true as this can be overcome by zero-shot learning.

Overall, the paper is very well written, in excellent English, and with clear argumentation. But the contents do not deliver any new or original material. Rather, this is a reformulation of existing concepts using other terms. Some are by the way incorrectly used. The techno-concept is a conceptual model of the artifact. Figures included in the paper are actually various techniques from information visualization.

Round 2

Reviewer 1 Report

Thank you for thoroughly addressing my comments and providing a much improved version of the paper. In my opinion, your revised article meets the criteria for publication and makes a fine and innovative cross-disciplinary publication. Good job!